# Transcription tuned by S-nitrosylation underlies a mechanism for *Staphylococcus aureus* to circumvent vancomycin killing

Xueqin Shu [1], Yingying Shi [1], Yi Huang[1], Dan Yu [2] ✉ & Baolin Sun [1,3,4] ✉

Treatment of *Staphylococcus aureus* infections is a constant challenge due to emerging resistance to vancomycin, a last-resort drug. S-nitrosylation, the covalent attachment of a nitric oxide (NO) group to a cysteine thiol, mediates redox-based signaling for eukaryotic cellular functions. However, its role in bacteria is largely unknown. Here, proteomic analysis revealed that S-nitrosylation is a prominent growth feature of vancomycin-intermediate *S. aureus*. Deletion of NO synthase (NOS) or removal of S-nitrosylation from the redox-sensitive regulator MgrA or WalR resulted in thinner cell walls and increased vancomycin susceptibility, which was due to attenuated promoter binding and released repression of genes involved in cell wall metabolism. These genes failed to respond to $H_2O_2$-induced oxidation, suggesting distinct transcriptional responses to alternative modifications of the cysteine residue. Furthermore, treatment with a NOS inhibitor significantly decreased vancomycin resistance in *S. aureus*. This study reveals that transcriptional regulation via S-nitrosylation underlies a mechanism for NO-mediated bacterial antibiotic resistance.

*S*taphylococcus aureus is a major human pathogen that causes diverse severe infections, ranging from superficial skin infections to life-threatening endocarditis and septicemia[1,2]. The emergence of multi-drug resistance has caused *S. aureus* infection a global concern, especially resistance to vancomycin, a drug of last resort[2–5]. A growing body of research focuses on the resistance mechanisms and antibiotic discovery[6–13]. Nitric oxide (NO), a toxic gas but an important signaling molecule, was first reported to be synthesized in eukaryotes by NO synthase (NOS) and plays an essential role in regulating physiological and immune functions[14,15]. S-nitrosylation, a posttranslational modification on cysteine thiol mediated by NO, has been found to drive a large part of the ubiquitous influence of NO on mediating cellular functions, such as cardiomyocyte dysfunction, inflammation and apoptosis, thereby providing a prototype of redox-based cellular

signaling mechanisms regulated by S-nitrosylation[16–19]. Furthermore, several gram-positive microorganisms, including *S. aureus*, possess a bacterial NO synthase (bNOS), which is highly homologous to the oxygenase domain of eukaryotic NOS, to generate NO endogenously[20,21]. bNOS was reported to play an important role in protecting bacteria against a wide spectrum of antibiotics[22,23], but the precise underlying mechanism remains elusive. The endogenous S-nitrosylation and redox sensors in bacteria are less well studied, except OxyR, which serves as a master regulator of S-nitrosylation in *Escherichia coli*[24]. A previous study used exogenous NO to treat *S. aureus* and found that approximately 5.8% of proteins encoded in the *S. aureus* genome underwent S-nitrosylation, revealing that AgrA is a direct target of host NO to inhibit toxin production[25]. In the present study, we demonstrated an endogenous NO-mediated S-nitrosylation

[1]Department of Oncology, The First Affiliated Hospital of USTC, Division of Life Sciences and Medicine, University of Science and Technology of China, Hefei, Anhui, China. [2]Laboratory of Dermatology, Beijing Pediatric Research Institute, Beijing Children's Hospital, Capital Medical University, Key Laboratory of Major Diseases in Children, Ministry of Education, National Center for Children's Health, Beijing, China. [3]CAS Laboratory of Innate Immunity and Chronic Disease, University of Science and Technology of China, Hefei, China. [4]Hefei National Laboratory for Physical Sciences at Microscale, Hefei, China. ✉e-mail: danyu@ccmu.edu.cn; sunb@ustc.edu.cn

regulatory mechanism for vancomycin-intermediate *S. aureus* (VISA) to circumvent vancomycin killing.

## Results

### S-nitrosylation is a prominent growth feature of *S. aureus*

S-nitrosylation patterns were analyzed in a clinically isolated VISA strain XN108[26], which has been identified as a staphylococcal cassette chromosome *mec* (SCC*mec*) type III, multilocus ST239, and spa type 037 methicillin-resistant *S. aureus* (MRSA), by liquid chromatography-mass spectrometry (LC−MS)/MS. A proteomic analysis identified 484 unique S-nitrosocysteine residues in 302 proteins in 10.1% of the total proteins encoded in the *S. aureus* genome (Supplementary Data 1). The dataset included constituents of proteins with roles in some central processes, such as energy production and conversion, translation and ribosomal biogenesis, transcription and regulators, nucleotide transport and metabolism, as well as a few proteins in less represented pathways (Fig. 1a). These data demonstrated that S-nitrosylation is a prominent feature of growth in *S. aureus*.

### Identification of the transcription factor MgrA as a target of S-nitrosylation

Bacteria have evolved transcription factors to sense the redox status of the environment and to respond by adjusting gene expression[24,27]. Here, we observed that a major proportion (8.9%) of the S-nitrosylated proteins in *S. aureus* were involved in transcription; this included some classic regulators, such as MgrA, WalR, SarR, SarS, and ArcR (Fig. 1b). These results suggested the existence of a transcriptional regulatory network driven by S-nitrosylation to balance nitrosative stress in *S. aureus*. We therefore focused on transcriptional regulators in the mass spectrometry dataset. MgrA, a global regulatory protein involved in autolytic activity, multidrug resistance, and virulence[28–31] was found to be S-nitrosylated at the unique cysteine residue Cys12 (Fig. 1c). This residue is within the MarR-type helix-turn-helix (HTH) DNA binding domain and located at the interface of the protein dimer[30] (Fig. 1d, e). This was consistent with the results of a previous study, which identified MgrA Cys12 as an S-nitrosylation site through addition of extra NO to the *S. aureus* strain NCTC8325[25]. To further confirm MgrA as an NO target, an S-Nitrosylation Western Blot Kit (90105, Pierce) in which NO-modified thiols are replaced with a stable TMT label was employed. In consistent with the proteomic analysis, western blot result (Fig. 1f) showed that S-nitrosylated MgrA was obviously detected in the wild-type (WT) strain, but not in the NOS knockout (Δ*nos*) strain, indicating that S-nitrosylation formation of MgrA is dependent on the NO generated from NOS. Our results thus suggested that in the absence of exogenous NO, MgrA undergoes endogenous S-nitrosylation induced by NO generated from NOS.

### NO-mediated MgrA S-nitrosylation promotes vancomycin resistance in VISA

MgrA was reportedly involved in the regulation of vancomycin resistance[30,32,33]. Next, we investigated whether S-nitrosylation of MgrA affected the vancomycin resistance of VISA. Allelic replacement was performed to generate the *mgrA*C12S mutant strain by replacing the cysteine residue with a serine, which cannot be S-nitrosylated. The minimum inhibitory concentration (MIC) of vancomycin was evaluated in the WT and *mgrA*C12S mutant strains. This was conducted using the broth microdilution method in Mueller-Hinton II (MH II) medium following criteria set by the Clinical and Laboratory Standards Institute (CLSI). We observed a significant decrease in the MIC of vancomycin in the *mgrA*C12S mutant strain, from 8 μg/mL to 4 μg/mL, while the parent VISA strain grew well under 6 μg/mL vancomycin (Fig. 2a). This suggested that MgrA S-nitrosylation is important for the modulation of vancomycin susceptibility. To test the effect of endogenous NO on vancomycin resistance in *S. aureus*, the growth rate was compared between the WT and Δ*nos* strain in vancomycin-containing medium. The Δ*nos* strain showed a growth defect in medium containing 4 μg/mL vancomycin (Fig. 2a). The susceptibility of the WT and mutant strains to vancomycin was further evaluated with plate-sensitivity assay. Both mutants exhibited significant susceptibility to vancomycin in a concentration-dependent manner on plates containing vancomycin, whereas the WT strain displayed resistant growth under the same conditions (Fig. 2b). The occurrence of VISA is usually accompanied by a decrease in sensitivity to daptomycin and teicoplanin[34], we thus also measured daptomycin or teicoplanin MIC of the WT and Δ*nos* mutant. Similar to the results what we observed from vancomycin, the Δ*nos* mutant showed decreased MIC in medium containing daptomycin or teicoplanin (Supplementary Fig. 1). These results suggested a strong dependence of *S. aureus* resistance on the endogenous NO level.

The NOS inhibitor $N^G$-nitro-L-arginine methyl ester (L-NAME)[35,36] was next used to investigate whether inhibition of NOS activity could affect the vancomycin resistance of *S. aureus*. When 2 μg/mL vancomycin was added, the WT strain showed a slight decrease in growth rate, which was further inhibited by addition of 40 mM L-NAME (Fig. 2c). This indicated a strong dependence of *S. aureus* on NO levels for vancomycin resistance. To determine whether this effect was generalizable, the VISA strain Mu50 was also treated with L-NAME. We found a similar inhibitory effect of L-NAME on Mu50 as observed in XN108 (Fig. 2d). To demonstrate that the growth inhibition effect of L-NAME was not due to L-NAME itself, it was used to treat Δ*nos* bacteria. We hypothesized that if the growth inhibition observed in Fig. 2c was due to nonspecific activity of L-NAME, treatment of the Δ*nos* strain with L-NAME would also induce significant growth inhibition. However, L-NAME did not inhibit Δ*nos* growth. In fact, it showed a small growth stimulation effect, whereas a growth inhibition was found in the WT when added with 40 mM L-NAME (Supplementary Fig. 2a, b). This is possibly due to the fact that L-NAME is an L-Arginine analog and bacteria may use it for growth when NOS is absent in cells. Thus, the results suggested that the growth inhibition caused by L-NAME was a result of NOS activity inhibition rather than an effect nonspecific to L-NAME. Together, these data demonstrated that VISA requires S-nitrosylation of MgrA to regulate vancomycin resistance, which is dependent on endogenous NO.

### MgrA S-nitrosylation facilitates vancomycin resistance in a NO-dependent manner

We next explored whether NO addition could compensate the defective growth of Δ*nos* or *mgrA*C12S in vancomycin-containing medium. Growth medium containing 2 μg/mL vancomycin was supplemented with the exogenous NO donor sodium nitroprusside (SNP). The addition of 1 or 5 μM SNP, but not 20 μM, could restore the decreased growth of Δ*nos* to some extent (Fig. 3a–c). This suggested a protective role of low-concentration NO in the defense of *S. aureus* against vancomycin. In contrast, the *mgrA*C12S growth rate did not recover when SNP was added (Fig. 3d–f). Moreover, we observed that the addition of SNP caused an obvious and dose-related inhibitory effect on the growth of both the WT and *mgrA*C12S strains (Fig. 3d–f), with a much stronger inhibitory effect on *mgrA*C12S. This indicated that the lack of MgrA S-nitrosylation severely impaired the ability of *S. aureus* to defend against excessive nitrosative stress. These results indicated that low-concentration NO and the induction of MgrA S-nitrosylation were required for *S. aureus* vancomycin resistance.

### S-nitrosylation of MgrA contributes to cell wall thickness and integrity

VISA strains typically show thickened cell walls, which allow increased capture of vancomycin molecules by free D-Ala−D-Ala residues, protecting cells from vancomycin killing[37]. The role of MgrA S-nitrosylation in determining cell wall thickness was therefore

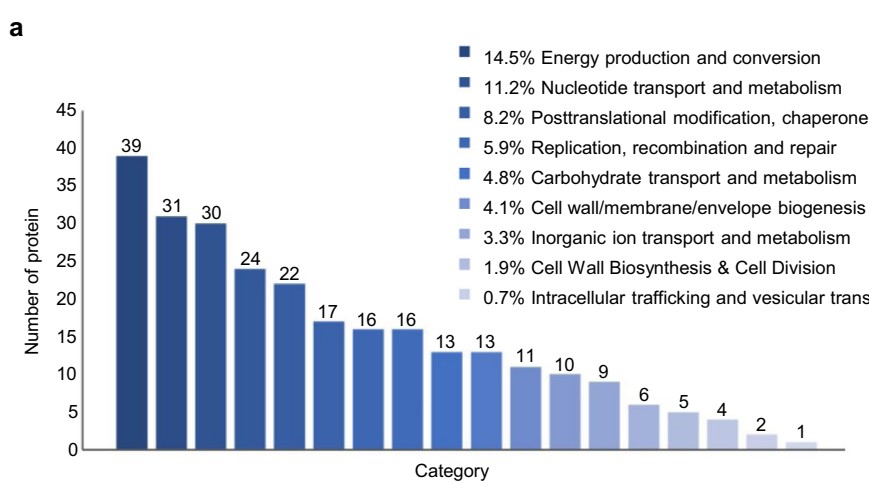

**a**

| | |
|---|---|
| 14.5% Energy production and conversion | 11.5% Translation, ribosomal biogenesis |
| 11.2% Nucleotide transport and metabolism | 8.9% Transcription & Regulators |
| 8.2% Posttranslational modification, chaperones | 6.3% Amino acid transport and metabolism |
| 5.9% Replication, recombination and repair | 5.9% Coenzyme transport and metabolism |
| 4.8% Carbohydrate transport and metabolism | 4.8% Lipid transport and metabolism |
| 4.1% Cell wall/membrane/envelope biogenesis | 3.7% Signal transduction mechanisms |
| 3.3% Inorganic ion transport and metabolism | 2.2% Secondary metabolites biosynthesis |
| 1.9% Cell Wall Biosynthesis & Cell Division | 1.5% Defense mechanisms |
| 0.7% Intracellular trafficking and vesicular transport | 0.4% Cell motility |

**b**

| Gene designation | Protein ID | Modified sequence | Position | Protein description |
|---|---|---|---|---|
| *mgrA* | AID39196.1 | EQLC(io)FSLYNAQR | C12 | Transcriptional regulator MgrA |
| *walR* | AID38461.1 | DGMEVC(io)R | C67 | Transcriptional regulatory WalR |
| *sarR* | AID40984.1 | C(io)SEFKPYYLTK | C57 | Transcriptional regulator SarR |
| *sarS* | AID38538.1 | IVSDLC(io)YK | C61 | HTH-type transcriptional regulator SarS |
| *arcR* | AID41343.1 | HLFNDVC(io)V | C233 | HTH-type transcriptional regulator ArcR |
| *rsbU* | AID40753.1 | GLIDESLTC(io)QDK | C18 | Sigma factor B regulation protein |
| *troR* | AID39141.1 | LDSLLNFPETC(io)PHGGVIPR | C124 | DtxR family transcriptional regulator |
| *ureG* | AID40982.1 | IIGVETGGC(io)PHTAIR | C69 | Urease accessory protein UreG |
| *yesN* | AID38639.1 | SLAPC(io)HDAFQPLLK | C135 | Putative response regulator |
| *mraZ* | AID39617.1 | EC(io)TVIGVSNR | C103 | Uncharacterized protein conserved in bacteria |
| *nusA* | AID39711.1 | ISVFSENNDIDAVGAC(io)VGAK | C251 | Putative N utilization substance protein A |
| *SAXN108_0312* | AID38753.1 | SEVDIYGC(io)ASAGIGER | C129 | Phage repressor |
| *SAXN108_2103* | AID40544.1 | TPVNVNGC(io)VSAGVGER | C128 | Phage repressor |
| *SAXN108_1765* | AID40206.1 | NGLQLGDTLNC(io)SGAESYK | C216 | LacI family transcriptional regulator |

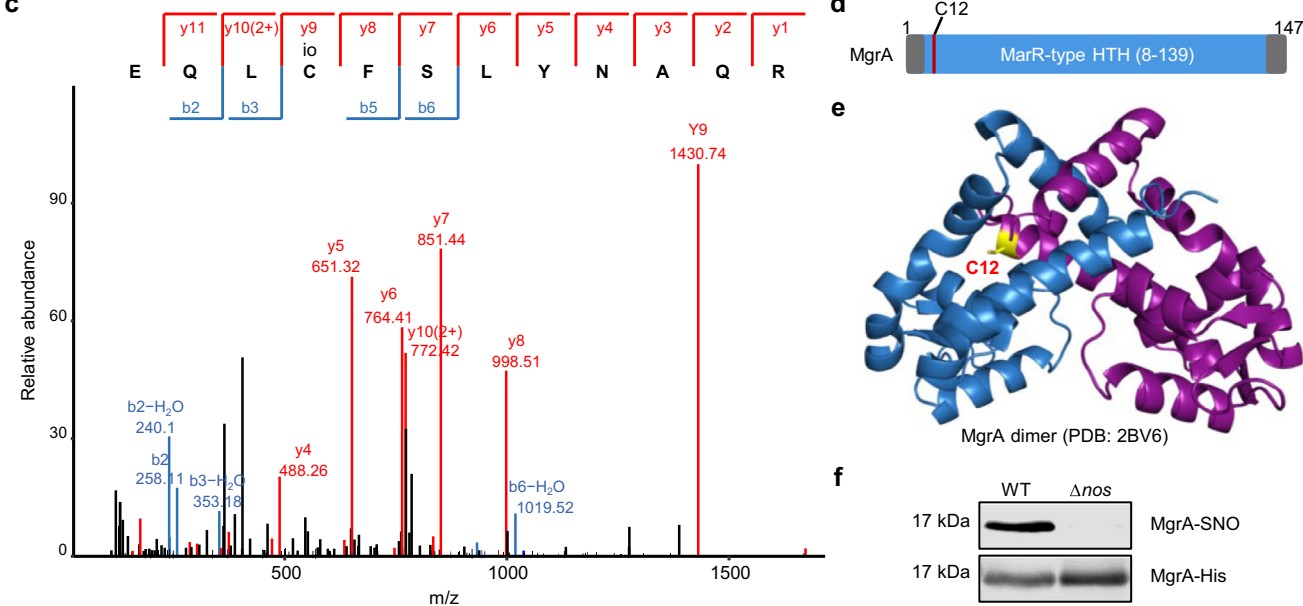

**Fig. 1 | Identification of the transcription factor MgrA as a target of S-nitrosylation. a** Summary of MS identified NO-modified proteins in *S. aureus* categorized by biological pathway. **b** Transcriptional regulators identified to be S-nitrosylated are listed. **c** Higher-energy collision dissociation mass spectrum of the cystine 12-containing peptide from MgrA. S-nitrosylated cysteine was labeled with isobaric iodoTMT and marked with io. The cysteine residue Cys12 of MgrA is located within the MarR-type HTH domain (**d**) and located at the interface of MgrA dimer (**e**). **f** S-nitrosylation of MgrA purified from the XN108 WT or Δ*nos* strain was detected by western blot. S-nitrosylated MgrA (MgrA-SNO) was detected with an anti-TMT antibody. Total MgrA (MgrA-His) was detected with an anti-His antibody. Data are representative of *n* = 3 biological replicates. Source data are provided as a Source Data file.

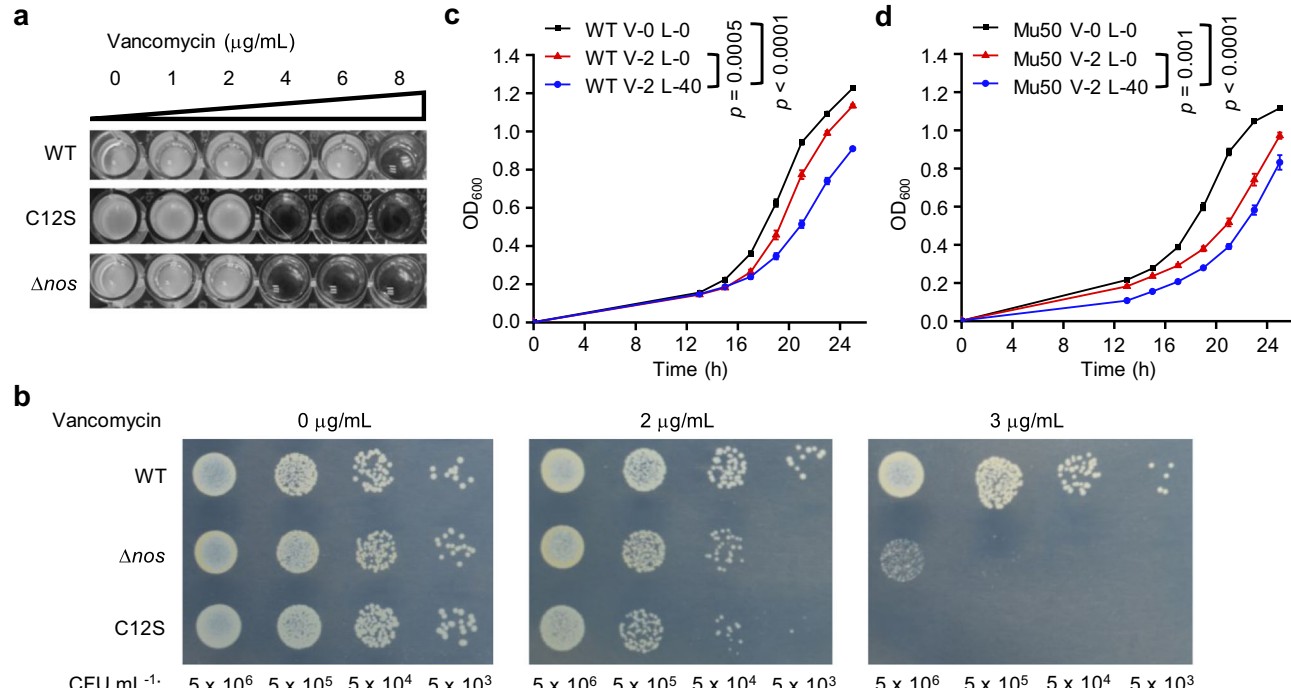

**Fig. 2 | NO-mediated MgrA S-nitrosylation promotes the resistance of VISA to vancomycin. a** MICs were assessed after the WT *S. aureus* XN108, *mgrA*C12S mutant and Δ*nos* mutant strains were grown under vancomycin conditions in a 96-well plate with shaking at 200 rpm for 48 h at 37 °C. Data are representative of $n = 3$ biological replicates. **b** The growth of all the strains on normal condition or medium with different concentrations of vancomycin as indicated on agar plates was measured by a plate-sensitivity assay. Data are representative of $n = 3$ biological replicates. CFU colony-forming units. Growth curve of the WT VISA strain XN108 (**c**) or Mu50 (**d**) in normal MH II medium (V-0 L-0) or medium containing 2 µg/mL vancomycin (V-2 L-0) or supplemented with 40 mM L-NAME (V-2 L-40) in 96-well plates with an initial OD$_{600}$ of 0.001. These cultures were incubated at 37 °C without shaking overnight (13 h) and then grown with shaking at 200 rpm for 12 h. Data are means of $n = 3$ biological replicates with SD. Two-sided two-way ANOVA, ***$p \le 0.001$; ****$p \le 0.0001$. Source data are provided as a Source Data file.

investigated via transmission electron microscopy (TEM). The cell wall of the WT strain exhibited an average thickness of $61.35 \pm 2.96$ nm, whereas the *mgrA*C12S mutant had a significantly thinner cell wall ($44.93 \pm 2.87$ nm, $p < 0.0001$). The Δ*nos* cells also had decreased cell wall thickness ($51.57 \pm 1.84$ nm, $p = 0.0001$), but the difference from the WT was not as visually apparent as that of the *mgrA*C12S mutant strain (Fig. 4a, b). Therefore, these results demonstrated that S-nitrosylation of MgrA Cys12 is important for maintaining uniform cell wall thickness.

To determine the mechanism by which MgrA S-nitrosylation affected cell wall thickness, we first investigated whether regulation of cell wall synthesis was involved. Quantitative reverse transcription (qRT)-PCR was performed in the WT and mutant strains to examine the transcriptional levels of genes that have previously been reported to regulate cell wall synthesis. However, most of these targets showed negligible changes in the *mgrA*C12S or Δ*nos* mutant strains compared to the WT strain (Supplementary Fig. 3), indicating that NO-mediated S-nitrosylation of MgrA did not specifically induce signaling change in the pathway associated with cell wall synthesis.

MgrA reportedly regulates transcription of genes involved in cell wall metabolism and autolysis, which is important for cell wall turn-over and integrity[38–40]. Thus, we measured the autolytic activity of the WT and mutant strains induced by 0.2% (v/v) Triton X-100 in Tri-HCl buffer. The *mgrA*C12S mutant strain showed a significantly faster rate of autolysis than the WT strain. Consistent with this observation, the Δ*nos* mutant also exhibited an increased autolysis rate (Fig. 4c, d). To examine the effect of NO on autolysis, SNP was added to Tri-HCl buffer containing WT, Δ*nos*, or *mgrA*C12S bacteria. We observed that the increased autolysis rate of the Δ*nos* mutant strain could be rescued by SNP, but SNP could not rescue the autolysis activity of *mgrA*C12S, and had no influence on the WT strain (Fig. 4c, d). This result was consistent with our above notion that NO could

compensate the growth inhibition of the Δ*nos* mutant in medium containing vancomycin. These results suggested that NO-mediated S-nitrosylation of MgrA played an important role in the maintenance of cell wall thickness and integrity, which is primarily achieved through altering autolytic activity.

## S-nitrosylation of MgrA drives transcriptional signaling through autolysis-related genes

We then explored whether the effect of MgrA S-nitrosylation on *S. aureus* autolysis resulted from functional control of autolysis-associated gene expression. qRT-PCR analyses were therefore performed in the WT and *mgrA*C12S mutant strains to measure the expression of *sarV*, *lytN*, and *altA*[38,41]. The results showed that the transcriptional levels of *lytN* and *sarV*, but not *altA*, were significantly upregulated in the *mgrA*C12S mutant strain compared to the WT strain (Fig. 4e). This was consistent with previous findings that MgrA acts as a repressor of autolysis genes in a direct or indirect manner[31,42], suggesting that MgrA S-nitrosylation negatively regulated autolysis in *S. aureus* by promoting transcriptional repression of genes involved in autolysis.

Based on our finding that NO can compensate the decreased growth rate of VISA in medium containing vancomycin, we investigated whether NO conveyed the signal for vancomycin resistance through transcriptional regulation by MgrA. The WT and *mgrA*C12S mutant strains were treated with 0.2 or 1 mM SNP. qRT-PCR analyses showed that addition of either 0.2 or 1 mM SNP significantly downregulated both *lytN* and *sarV* in the WT strain. In contrast, there was no noticeable change in these genes in the *mgrA*C12S mutant strain. This suggested that removal of the S-nitrosylation from MgrA Cys12 severely impaired NO-mediated signal transduction for transcriptional regulation of *lytN* and *sarV* (Fig. 4f).

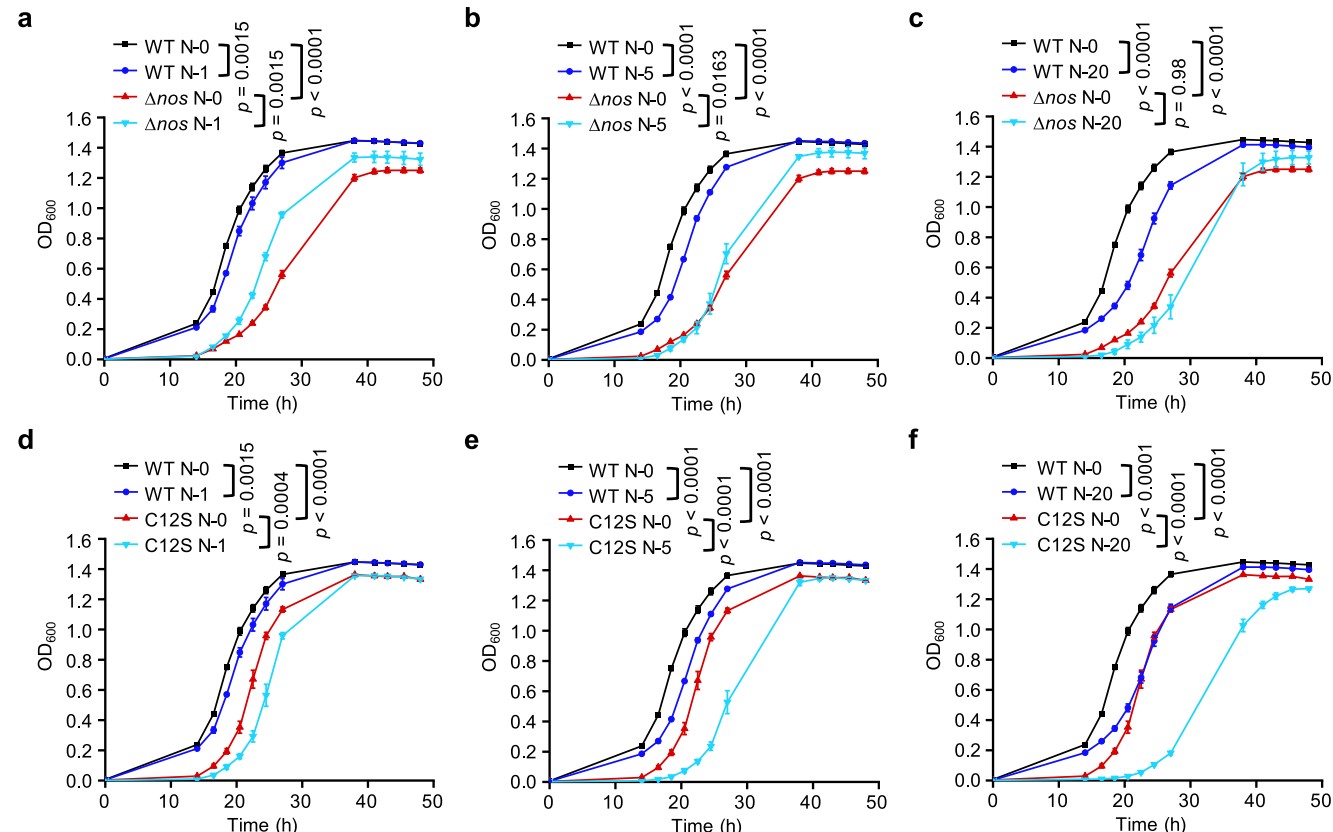

**Fig. 3 | NO is required for S-nitrosylation on MgrA to facilitate vancomycin resistance.** The WT and Δ*nos* strains (**a**–**c**) or the WT and *mgrA*C12S strains (**d**–**f**) were grown in MH II medium containing 2 μg/mL vancomycin in 96-well plates with shaking at 200 rpm for 48 h at 37 °C. OD$_{600}$ was monitored to evaluate the growth when different concentrations of SNP were added. N-0, N-1, N-5 or N-20 represents 0, 1, 5 or 20 μM SNP respectively. Data are means of $n = 3$ biological replicates with SD. Two-sided two-way ANOVA, *$p \leq 0.05$; **$p \leq 0.01$; ***$p \leq 0.001$; ****$p \leq 0.0001$; ns not significant. Source data are provided as a Source Data file.

Interestingly, we found that MgrA S-nitrosylation regulated the transcription of genes involved in autolysis, while a previous study showed that oxidation of MgrA at Cys12 likely induces signaling in a different regulatory pathway[30]. To determine whether S-nitrosylation and S-oxidation of this single residue in MgrA control distinct signaling pathways, H$_2$O$_2$ was used to oxidize Cys12 in MgrA, and the effects on *lytN* and *sarV* expression in the WT and *mgrA*C12S strains were measured. Compared to NO treatment, treatment with 0.05, 0.2, or 1 mM H$_2$O$_2$ all failed to decrease the transcription of *lytN* or *sarV* in either WT or *mgrA*C12S strains (Fig. 4g). This indicated that H$_2$O$_2$-mediated S-oxidation of MgrA and NO-mediated S-nitrosylation of MgrA likely drived different regulatory pathways. Distinct transcriptional responses controlled by alternative posttranslational modifications of a transcriptional regulator has also been observed in the global regulator OxyR in *E. coli*[24]; it may therefore be a general signal-delivery strategy used by bacteria to execute functions precisely and specifically.

**Removal of MgrA S-nitrosylation impairs its DNA binding in gene promoters**

How can MgrA S-nitrosylation affect gene transcriptional control? It has been reported that MgrA regulates gene transcription through association with gene promoters[40,42]. We reasoned that the contribution of MgrA S-nitrosylation to downstream gene regulation may be a result of its promoter binding capacity. Chromatin immunoprecipitation (ChIP)-qPCR was used to evaluate the effect of S-nitrosylation on MgrA occupancy at the *lytN* and *sarV* promoters in vivo. WT MgrA showed significant enrichment in these promoters, but the C12S mutant exhibited no obvious binding (Fig. 5a, b),

indicating a dependence of MgrA on S-nitrosylation to associate with the *lytN* and *sarV* promoters.

Interestingly, a previous study found that oxidation of MgrA Cys12 influences MgrA dimerization and DNA binding activity[30]. Thus, to verify that the observed decrease in DNA binding ability of MgrA was due to the loss of S-nitrosylation, we examined the effect of endogenous NO on the DNA binding ability of MgrA. MgrA occupancy was measured at the promoters of *lytN* and *sarV* in the WT and Δ*nos* strains. ChIP-qPCR results showed that NOS knockout caused an obvious decrease in MgrA binding to the *lytN* and *sarV* promoters (Fig. 5c, d). These results suggested that NO-mediated S-nitrosylation of MgrA at Cys12 affected its DNA binding ability in vivo, consequently interfering with gene transcription.

To further verify that S-nitrosylation affected the ability of MgrA to associate with DNA, we used a fluorescein amidite (FAM)-labeled probe containing the promoter fragment and compared the DNA binding affinity between the WT and C12S-mutant MgrA by performing gel-shift experiments in vitro. Consistent with the results in vivo, the WT MgrA showed a marked shift on the native gel, while loss of S-nitrosylation impaired the binding efficacy of MgrA with the *lytN* and *sarV* promoters (Fig. 5e, f). These data confirmed our proposed regulatory mechanism for MgrA and highlighted the importance of S-nitrosylation for the DNA binding ability of MgrA.

To provide additional evidence for the proposed regulatory mechanism, we constructed an *mgrA*C12A mutant in the WT and Δ*nos* strains with allelic replacement. We first measured the vancomycin MIC of these strains. Consistent with results from the *mgrA*C12S mutant, the *mgrA*C12A mutant also showed a decreased MIC at 4 μg/mL in

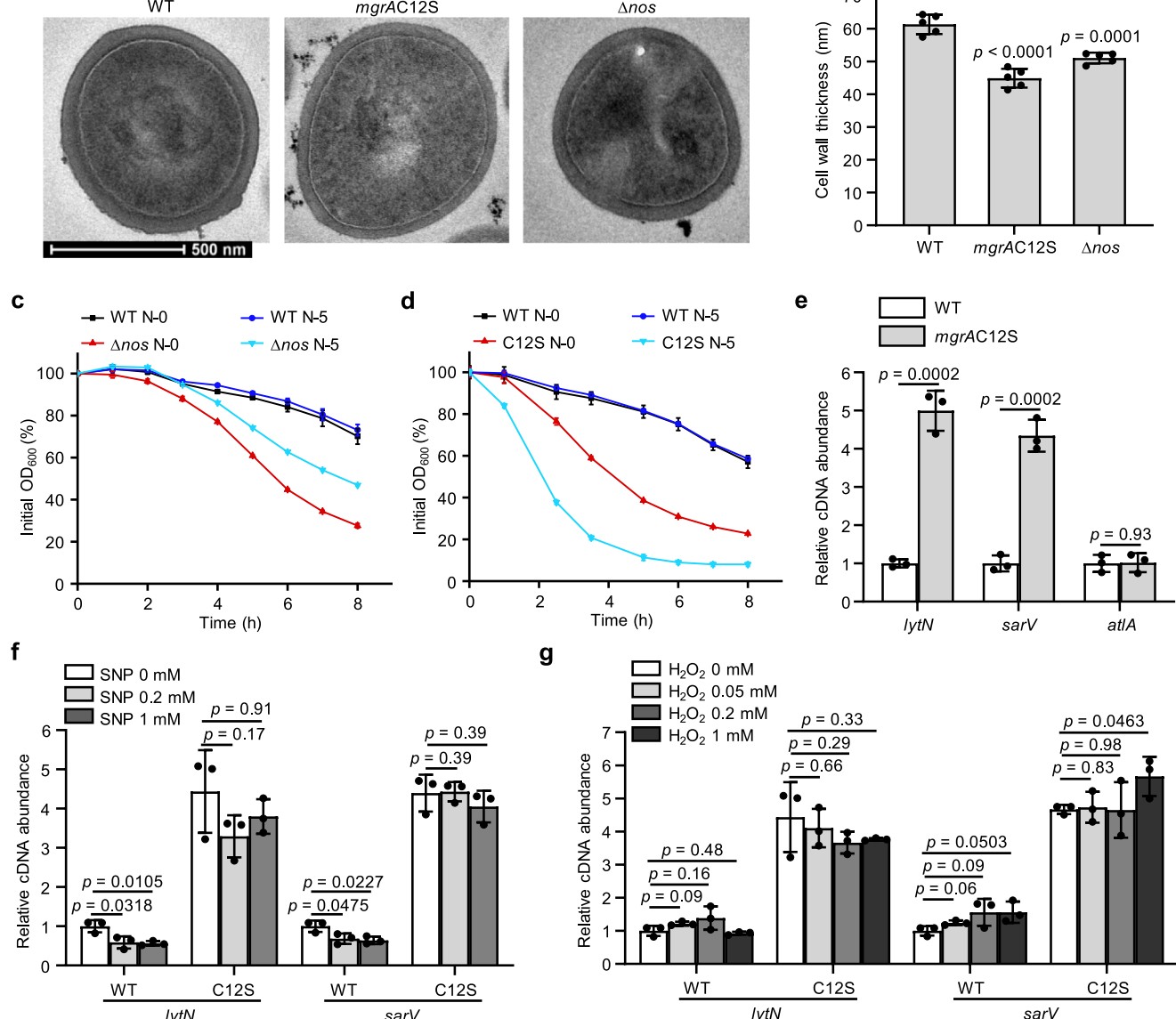

**Fig. 4 | Redox-signal transduction by MgrA to vancomycin resistance is achieved by regulating cell wall thickness and cell autolysis. a, b** Cell wall thickness was measured by transmission electron microscopy (TEM). Representative TEM images of the WT, *mgrA*C12S and Δ*nos* strains cultured with shaking at 220 rpm for 16 h at 37 °C are displayed (**a**). Scale bar = 500 nm. Cell wall thickness of $n = 5$ individual cells from each strain were measured three times randomly and expressed as means with SD (**b**). Triton X-100-induced autolysis of the WT and Δ*nos* strains (**c**) or the WT and *mgrA*C12S strains (**d**) was measured in Tri-HCl buffer supplemented with or without 5 μM SNP (displayed as N-5 or N-0). The percentage of the initial $OD_{600}$ was displayed. Data are means of $n = 3$ biological replicates with SD. **e** The transcriptional levels of the indicated genes that reported to regulate autolysis activity were tested in the WT and *mgrA*C12S mutant strains by using qRT-PCR. Data are means of $n = 3$ biological replicates with SD. The transcriptional levels of *lytN* and *sarV* were measured in both strains after incubation with different concentrations of SNP (**f**) or $H_2O_2$ (**g**) as indicated. Data are means of $n = 3$ biological replicates with SD. Two-sided unpaired Student's *t*-test, *$p \leq 0.05$; ***$p \leq 0.001$; ****$p \leq 0.0001$; ns not significant (**b, e, f, g**). Source data are provided as a Source Data file.

comparison to the WT strain (Supplementary Fig. 4). In addition, the *mgrA*C12A mutation in the *nos* knockout background did not further decrease the vancomycin MIC (Supplementary Fig. 4).

Then, the effect of NO donor on the growth of the *mgrA*C12A mutant was measured in MH II medium containing vancomycin. We found that addition of SNP (1, 5, or 20 μM) could not compensate the defective growth of the *mgrA*C12A mutant but exhibited a significant and dose-dependent inhibitory effect (Supplementary Fig. 5), which was consistent with our findings in the *mgrA*C12S strain.

The autolytic activity and transcription of genes that involved in autolysis in the *mgrA*C12A mutant were also measured. We found that the *mgrA*C12A mutant showed increased autolytic activity when compared with the WT strain, which is comparable to that of the *mgrA*C12S

mutant and could not be compensated by treatment with NO donor (Supplementary Fig. 6a). qRT-PCR analyses showed that *lytN* and *sarV* were upregulated in the *mgrA*C12A mutant compared with the WT strain (Supplementary Fig. 6b), which is also consistent with results in the *mgrA*C12S mutant strain. Furthermore, we treated the *mgrA*C12A mutant strains with 0.2 or 1 mM SNP as did in Fig. 4f. NO donor treatment did not induce significant changes in *lytN* and *sarV* expression in the *mgrA*C12A mutant strain (Supplementary Fig. 6c), which further confirmed our conclusion that removal of the S-nitrosylation from MgrA Cys12 severely impaired NO-mediated signal transduction to *lytN* and *sarV*. Collectively, these data demonstrated a role of S-nitrosylation at MgrA Cys12 in modulating vancomycin susceptibility in *S. aureus*.

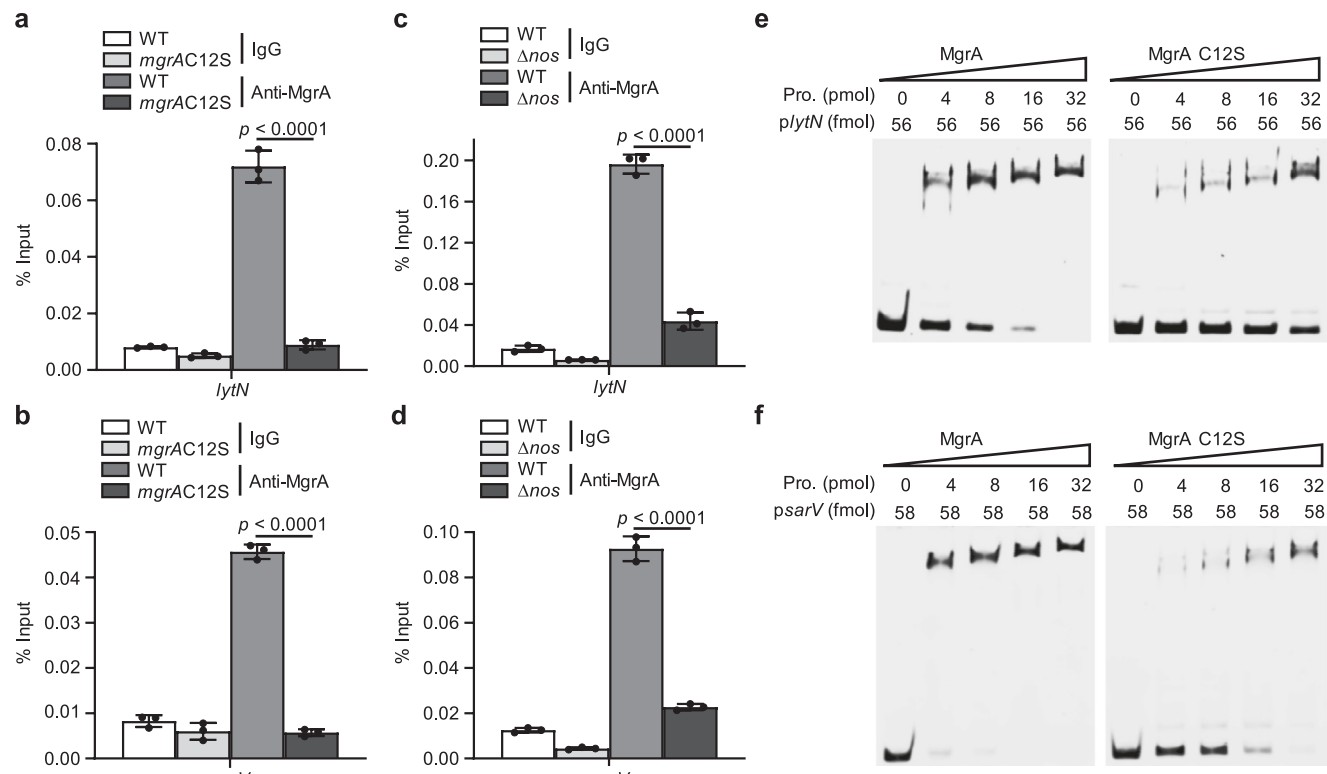

**Fig. 5 | Removal of MgrA S-nitrosylation impairs its DNA binding activity.** ChIP-qPCR was used to evaluate the occupancy of endogenous WT MgrA or MgrAC12S at the promoters of *lytN* and *sarV* (**a**, **b**), and the occupancy of endogenous WT MgrA in the WT or Δ*nos* strains (**c**, **d**). Data are means of *n* = 3 biological replicates with SD. Two-sided unpaired Student's *t*-test, ****$p \le 0.0001$. DNA binding efficacy of WT and C12S mutant MgrA at *lytN* (**e**) or *sarV* (**f**) promoters was compared by using gel-shift assay. Different concentrations of purified MgrA or MgrAC12S protein were incubated with a FAM-labeled probe containing the MgrA binding sequence on the *lytN* or *sarV* promoter region. The probe concentrations were indicated as p*lytN* or p*sarV*. The upper bands represent probes bound with protein; the lower bands represent free probes. Data are representative of *n* = 3 biological replicates. Source data are provided as a Source Data file.

## S-nitrosylation of WalR confers vancomycin resistance

Then, we wondered whether other S-nitrosylated transcriptional regulators may also contribute to vancomycin resistance in VISA and whether the S-nitrosylation-mediated mechanism may be generalizable to other regulators. WalR, the response regulator of the two-component regulatory system WalKR, which has been reported to regulate autolysis and cell wall metabolism in *S. aureus*[40], was identified as S-nitrosylated at Cys67 (Supplementary Fig. 7a). Cys67 is located in the receiver domain that accepts the signal from the cognate histidine kinase[43] (Supplementary Fig. 7b, c). S-nitrosylation of WalR was also detected by western blot, as shown in Supplementary Fig. 8, S-nitrosylated WalR was obviously detected in the WT strain, but not in the Δ*nos* strain, indicating that S-nitrosylation formation of WalR is dependent on the NO generated from NOS. To determine whether WalR S-nitrosylation modulated vancomycin resistance in *S. aureus*, a *walR*C67S substitution strain was generated and vancomycin MIC was measured. Similar to the results observed in the *mgrA*C12S mutant, the *walR*C67S mutant strain exhibited a significant reduction in the vancomycin MIC from 8 μg/mL to 4 μg/mL (Fig. 6a, b), which was consistent with the previous findings that WalR plays a role in regulating vancomycin resistance[44,45]. The *walR*C67S mutant strain also exhibited significant growth defects on plates containing vancomycin (Fig. 6c). These data indicated that NO-mediated S-nitrosylation of WalR also exerts an enhancement for VISA to exhibit vancomycin resistance.

A significant decrease in cell wall thickness was also observed in the *walR*C67S mutant strain ($54.87 \pm 4.04$ nm, $p = 0.0198$) (Fig. 6d, e). In addition, the role of S-nitrosylation in WalR function was determined by comparing the autolysis rate between the WT and *walR*C67S mutant strains. The *walR*C67S mutant strain displayed increased autolytic activity compared with the WT strain (Fig. 6f). qRT-PCR analyses showed that *sle1* and *atlA*, which are involved in autolysis and are reportedly regulated by WalR, were significantly upregulated in the *walR*C67S mutant strain (Fig. 6g). ChIP-qPCR analyses showed that NOS knockout also caused an obvious decrease in WalR binding to the *sle1* and *atlA* promoters (Fig. 6h). These results were consistent with our findings for MgrA, suggesting that redox-signal transduction through S-nitrosylation to gene transcription by these regulators is likely a general regulatory model in *S. aureus*.

To investigate the effects of MgrA and WalR double mutation on vancomycin resistance, we constructed an *mgrA*C12S/*walR*C67S double mutant. We found that this double mutant grew as well as the WT strain, which probably because that single mutation in either *mgrA* or *walR* did not cause growth defective. Rather, these mutants showed increased growth as compared to the parent strain (Supplementary Fig. 9a). The MIC of vancomycin was compared between the WT, *mgrA*C12S, *walR*C67S, and *mgrA*C12S/*walR*C67S strains. We found that the *mgrA*C12S/*walR*C67S mutant showed growth defect at 4 μg/mL vancomycin, with no significant difference compared to the *mgrA*C12S or *walR*C67S mutant (Supplementary Fig. 9b), which is consistent with the observation that the Δ*nos* strain retained a MIC of 4 μg/mL. This may be because although MgrA and WalR are both transcriptional regulators influencing similar genes involved in autolysin production, they are likely not collaborators, and thus do not have coordinated function and are not dependent on one another to regulate vancomycin resistance[31,46]. Besides, several genes, such as *vraTSR*, *graSR* and *rpoB*, have also been reported to contribute to vancomycin resistance in *S. aureus* and important for VISA development[47], which are likely independent of NO-mediated S-nitrosylation.

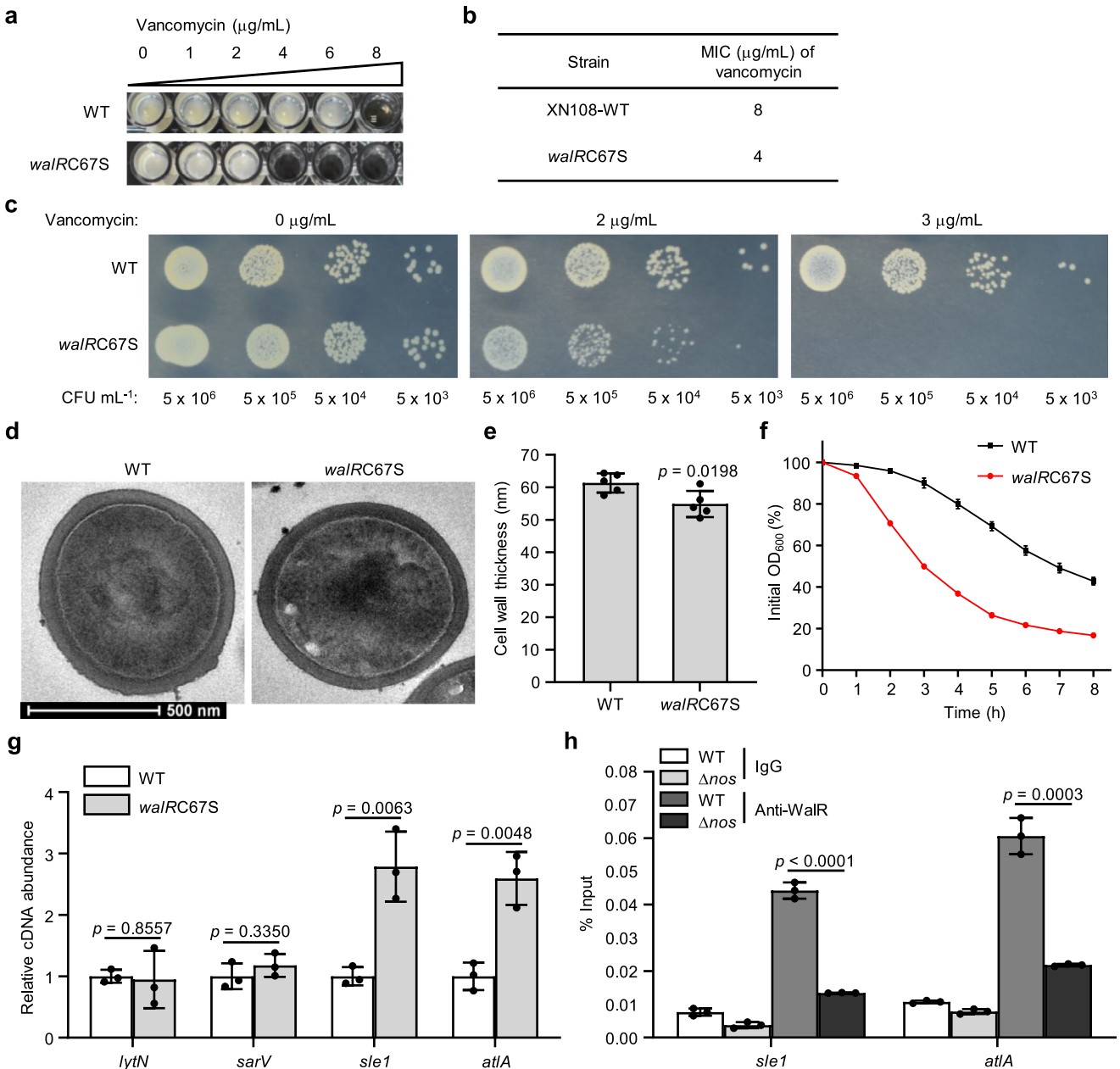

**Fig. 6 | WalR adopts S-nitrosylation to confer vancomycin resistance. a** MICs were assessed after the WT and *walR*C67S mutant strains were grown under vancomycin conditions in a 96-well plate with shaking at 200 rpm for 48 h at 37 °C. Data are representative of *n* = 3 biological replicates. **b** The difference in MICs observed in **a** is displayed. **c** The growth of both strains on normal condition or medium containing different concentrations of vancomycin as indicated on agar plates was measured by a plate-sensitivity assay. CFU colony-forming units. **d, e** Cell wall thickness was measured by transmission electron microscopy (TEM). Representative TEM images of the WT and *walR*C67S strains cultured with shaking at 220 rpm for 16 h at 37 °C are displayed (**d**). Scale bar = 500 nm. Cell wall thickness of *n* = 5 individual cells from each strain were measured three times randomly and

expressed as means with SD (**e**). **f** Triton X-100-induced autolysis of the WT and *walR*C67S mutant strains was measured in 96-well plate with shaking at 200 rpm for 8 h at 37 °C. The percentage of the initial $OD_{600}$ was displayed. Data are means of *n* = 3 biological replicates with SD. **g** The transcriptional levels of selected genes that were reported to regulate autolysis were tested in all the strains by using qRT-PCR. Data are means of *n* = 3 biological replicates with SD. **h** ChIP-qPCR was used to evaluate the occupancy of WT WalR or WalRC67S at promoters of *sle1* and *altA* in the WT and Δ*nos* strains. Data are means of *n* = 3 biological replicates with SD. Two-sided unpaired Student's *t*-test, *$p \leq 0.05$; **$p \leq 0.01$; ***$p \leq 0.001$; ****$p \leq 0.0001$; ns not significant (**e, g, h**). Source data are provided as a Source Data file.

## Discussion

In summary, this study demonstrated endogenous S-nitrosylation as a key mechanism for regulation of vancomycin resistance in *S. aureus* and as an important area of future bacterial research. VISA uses a mechanism involving transcriptional regulators (such as MgrA and WalR) to sense endogenous NO, causing S-nitrosylation and inducing a transcriptional signal to circumvent vancomycin killing (Fig. 7). This

study provides valuable insights for the clinical treatment of VISA and other antibiotic-resistant pathogens.

MgrA belongs to the MarR family regulators, which are widely distributed among bacteria and regulate diverse microbial physiological processes. Intriguingly, some other regulators, including SarA and SarZ in *S. aureus* and OhrR in *Bacillus subtilis*, also contain this conservative redox-sensitive cysteine residue as that in MgrA

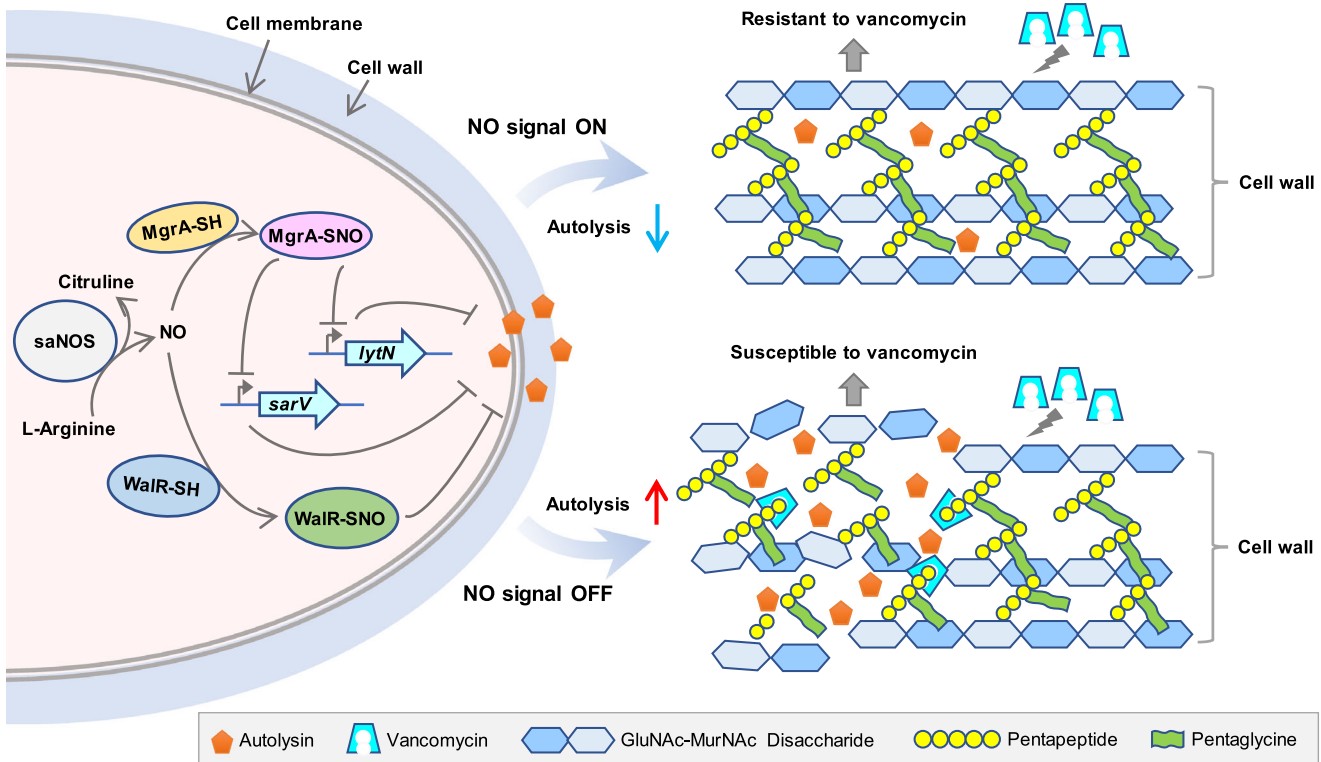

**Fig. 7 | Mechanism of S-nitrosylation signal transduction to vancomycin resistance.** *S. aureus* has evolved saNOS to generate NO. When the NO signal is on, MgrA and WalR undergo S-nitrosylation on the cysteine thiol, thereby initiating the transcriptional repression signals to the downstream genes such as *lytN* and *sarV* to inhibit autolysin expression, which leads to a thickened cell wall and increased resistance to vancomycin. When the NO signal is off, the autolysin production will be increased, thus resulting in accelerated cell wall degradation and increased susceptibility to vancomycin. The *lytN* gene encodes an autolysin protein. The *sarV* gene encodes a SarV protein, which is a positive regulator of autolysin genes. GlcNAc, *N*-acetylglucosamine. MurNAc, *N*-acetylmuramic acid. Pentapeptide, L-Ala−D-Gln−L-Lys−D-Ala−D-Ala.

(Supplementary Fig. 10). This indicates that S-nitrosylation may affect antibiotic resistance or other cellular functions through a similar mechanism in other bacterial species.

The S-nitrosylated cysteine in MgrA is critical for transcriptional activation of downstream genes that are involved in cell autolysis. S-oxidation of the same cysteine residue has been found to initiate a distinct regulatory signal. It is important to note that this unique cysteine in MgrA is sometimes phosphorylated[48], and oxidation of Cys12 in the *S. aureus* strain Newman causes antibiotic resistance[30], suggesting that *S. aureus* uses modification of Cys12 as a versatile mechanism for responding to cellular nitrosative or oxidative stress and defending against antibiotics. Future studies will uncover other S-nitrosylation-mediated signaling pathways in bacteria.

NOS has also been reported to influence *S. aureus* virulence[23,49], sensitivity to oxidative stress[23,35] and heme stress[50]; it also plays essential roles in regulating electron transfer to maintain membrane bioenergetics[36] and modulates aerobic respiratory metabolism and cell physiology[51]. Bacterial NOS-derived NO also plays important roles in regulating the formation, swarming, and dispersal of bacterial biofilms[52,53]. These findings suggest a key role of bacterial NOS or NOS-derived NO in the global regulation of *S. aureus* physiological phenotypes. It is thus worth exploring the mechanisms by which NOS is regulated. The regulatory mechanism mediated by S-nitrosylation may be universal and could be applied to regulation of other physiological phenotypes, and it should therefore be further explored in the future.

Here, we found that the NOS inhibitor L-NAME and vancomycin showed a synergistic antibacterial effect on *S. aureus*. Introducing a NOS inhibitor as a co-treatment with other antibiotics is thus a potential strategy for clinical treatment of severe *S. aureus* infection.

However, L-NAME is also an effective inhibitor of mammalian NOS[54], and further research is therefore required to discover a bacteria-specific NOS inhibitor.

Host-derived NOS and NO have been suggested to have roles in controlling *S. aureus* infections[25,55–60]. NO produced by host immune cells has been shown to have broad activity against diverse pathogens and to inhibit *Staphylococcus* virulence[15,25]. However, it has been found that NO produced by bacteria promotes bacterial virulence and drug resistance[35]. Thus, it would be useful to understand whether host cell-produced NO could affect bacterial antibiotic susceptibility. The present study provides a key insight that bacteria may use host-produced NO to resist antibiotic killing during infection. This is a promising direction for future study and potential improvement of antibiotic therapeutic strategies.

## Methods

### Bacterial strains, plasmids, and growth conditions

The bacterial strains and plasmids used in this study are listed in Supplementary Table 1. XN108 is a clinical VISA. *S. aureus* and *E. coli* strains were stored in 15% glycerol broth at −80 °C and subcultured before being used for any experiment. *S. aureus* strains were grown with shaking at 220 rpm in tryptic soy broth (TSB; 211825, Difco) medium or on TSA (TSB added with 15 g agar A per liter) plates at 37 °C. *E. coli* strains were grown with shaking at 220 rpm in LB medium (10 g tryptone, 5 g yeast extract and 5 g NaCl per liter) medium or on LA (LB added with 15 g agar A per liter) plates at 37 °C. When required, different antibiotics were added, 150 µg/mL ampicillin sodium salt or 50 µg/mL kanamycin sulfate for *E. coli* and 15 µg/mL chloramphenicol for *S. aureus* strains were supplemented in medium.

## Primers

Primers used in this study are listed in Supplementary Table 2.

## Construction of mutant *S. aureus* strains

To create mutant strain, the temperature sensitive plasmid pBTs was used as previously described[61]. The DNA fragment contains the upstream and downstream of the mutation site were amplified from XN108 wide-type strain genomic DNA. Primer pair of *nos*-up-F-KpnI/*nos*-up-R and *nos*-dn-F/*nos*-dn-R-SacI were used for *nos* knockout, primer pair of *mgrA*(C12S)-up-F-KpnI/*mgrA*(C12S)-up-R and *mgrA*(C12S)-dn-F/*mgrA*(C12S)-dn-R-SalI were used for *mgrA*(C12S) point mutation and primer pair of *walR*(C67S)-up-F-EcoRI/*walR*(C67S)-up-R and *walR*(C67S)-dn-F/*walR*(C67S)-dn-R-KpnI were used for *walR*(C67S) point mutation. The fragments were digested with indicated endonuclease (Thermo Scientific) and cloned into the plasmid pBTs with T4 ligase (EL0013, Thermo Scientific). The resulting plasmid was first introduced into *E. coli Trans*1-T1 (cultured at 37 °C) for replication, and then transformed into *S. aureus* strain RN4220 for modification and subsequently transformed into *S. aureus* strain XN108. Then, allelic replacement mutant was selected. The overnight cultures of plasmid containing *S. aureus* in TSB added with 15 µg/mL chloramphenicol at 30 °C were diluted 1:200 into TSB without antibiotics, and cultured at 42 °C for induction of homologous recombination. Every generation of the strain cultured at 42 °C were picked out and cultured on TSA plates containing 15 µg/mL chloramphenicol. Single clones were picked out and cultured in TSB without antibiotics at 30 °C for two generations for plasmid loss. The cultures were diluted and cultured on TSA plates containing 100 ng/mL anhydrotetracycline. Single clones were picked out and cultured in TSB without antibiotics and then in TSB containing 15 µg/mL chloramphenicol at 37 °C in 96-well plates. The clones that can only grow in TSB without antibiotics will be further confirmed by PCR and sequencing for the identification of allelic replacement mutants.

## Construction of protein expression plasmids

To construct the protein expression plasmids pETMgrA and pETM-grA(C12S), the full-length *mgrA* and *mgrA*C12S ORFs were amplified from XN108 wild-type and *mgrA*C12S mutant strain genomic DNA respectively using primer pairs of MgrA-ex-F-BamHI/MgrA-ex-R-SalI. The DNA fragments were digested with indicated endonuclease and then cloned into the expression vector pET28a (+). The recombinant expression plasmids were transformed into *E. coli* BL21 (DE3).

To construct the protein expression plasmid pLIMgrA-His, the full-length *mgrA* ORF was amplified from XN108 wild-type strain genomic DNA using primer pairs of ex-pLI-MgrA-His-F-EcoRI/ex-pLI-MgrA-His-R-BamHI. To construct the protein expression plasmid pLIWalR-His, the full-length *walR* ORF was amplified from XN108 wild-type strain genomic DNA using primer pairs of ex-pLI-WalR-His-F-EcoRI/ex-pLI-WalR-His-R-BamHI. The fragments were digested with indicated endonuclease and cloned into the plasmid pLI50. The resulting plasmid was first introduced into *E. coli Trans*1-T1 for replication, and then transformed into *S. aureus* strain RN4220 for modification and subsequently transformed into XN108 wild-type and Δ*nos* strains.

## Antibiotic sensitivity assays

The MICs of vancomycin were determined using a microdilution technique according to NCCLS guidelines in Mueller-Hinton II (MH II; 212322, Difco) medium. Overnight cultures of *S. aureus* strains were 1:100 diluted into fresh TSB and grown to mid-exponential phase before the number of bacterial cells were normalized and diluted to approximately $5 \times 10^5$ CFU (colony-forming units) per milliliter in MH II medium with gradient concentrations of vancomycin. The plates were incubated at 37 °C with shaking for 24–48 h before being visualized.

The plate-sensitivity assay was performed with fresh cultures grown to an $OD_{600}$ of 0.5 in TSB. The number of bacterial cells from all strains were normalized to approximately $5 \times 10^6$ CFU mL$^{-1}$ with PBS, followed by three 10-fold serial dilutions. Then 5 µL of each bacterial sample was spotted onto the LA plates containing gradient concentrations of vancomycin. All plates were incubated at 37 °C for 24 h.

## Transmission electron microscopy (TEM)

To detect morphological changes of the mutants, a transmission electron microscopy (TEM) (FEI) was used. Samples were prepared as previously described[62]. One milliliter of overnight cultured strains were harvested by centrifugation at $5000 \times g$ for 5 min, washed twice with 1 mL PBS buffer (pH 7.2), resuspended in 1 mL 2.5% (v/v) glutaraldehyde and incubated overnight at 4 °C. Then glutaraldehyde-fixed samples were washed three times with 1 mL PBS buffer with soft shaking for 20 min, and postfixed in 2.0% (w/v) osmium tetroxide overnight at room temperature. Samples were washed three times with 1 mL PBS buffer, dehydrated through an ethanol series (l mL 30%, 50%, 70%, 80%, 90% and 100% ethanol) and washed three times with 1 mL acetone. Samples were embedded in 500 µL LR White resin series once with the acetone/LR White resin ratio of 1:1 for 1 h, 1:3 for 3 h, and twice with total LR White resin for 6 h at room temperature. Samples were picked out into a new tube added with 250 µL LR White resin, embedded overnight at room temperature and then polymerized at 70 °C for at least 15 h. Samples were thin sectioned and observed with TEM operated at an accelerating voltage of 120 kV.

## Triton X-100-induced autolysis assay

Overnight-grown bacterial cells were diluted to an $OD_{600}$ of 0.05 in TSB and grown to an $OD_{600}$ of 0.8 at 37 °C. Cells were harvested by centrifugation at $5000 \times g$ for 3 min and washed twice with 50 mM Tris-HCl buffer (pH 7.5). Pellets were resuspended in the same buffer containing 0.2% (v/v) Triton X-100 and incubated at 37 °C with shaking at 180 rpm. The lysis was checked by measuring the progressive decrease in absorbance ($OD_{600}$) at one-hour intervals by using a microplate reader (Elx800, Bio-Tek).

## Total RNA extraction and qRT-PCR

The overnight cultures of *S. aureus* were diluted to an $OD_{600}$ of 0.05 in fresh TSB and cultivated to mid-exponential phase ($OD_{600} = 1$). The cells were collected by centrifugation at $12,000 \times g$ for 1 min and then immediately treated with 0.9 mL of RNAiso plus (108-95-2, TaKaRa) and lysed with 0.1-mmdiameter-silica beads in a FastPrep-24 automated system (MP Biomedicals). The residual DNA removal and cDNA synthesis were achieved by using a PrimeScript RT reagent kit with gDNA eraser (RR047A, TaKaRa). qRT-PCR was performed with SYBR Ex Taq premix (RR420A, TaKaRa) using the StepOne real-time PCR system (Applied Biosystems). The cDNA quantity measured by real-time PCR was normalized to the abundance of *pta* cDNA[63]. All qRT-PCR assays were repeated at least three times.

## Protein expression and purification

The BL21 (DE3) strain, containing the protein expression plasmid pETMgrA or pETMgrA(C12S), was grown in LB with 50 µg/mL kanamycin at 37 °C to an $OD_{600}$ of 0.6−0.8 and induced with 0.5 mM isopropyl-β-D-1-thiogalactopyranoside (IPTG) at 16 °C overnight. Cells were harvested by centrifugation and proteins were purified with a His-tag Protein Purification Kit (P2226, Beyotime) following the manufacturer's recommendation. Protein purity was analyzed by 15% SDS-PAGE, and followed by a determination of protein concentration using a Bradford Protein Assay Kit (P0006C, Beyotime) with bovine serum albumin as the standard.

## S-nitrosylated protein detection by western blot

S-nitrosylated protein was detected using an S-Nitrosylation Western Blot Kit (90105, Pierce) with some modification referring to a previous study[25]. Protein samples were protected from light until labeling

reagent addition. The WT *S. aureus* XN108 and Δ*nos* strains with pLIMgrA-His or pLIWalR-His plasmid were constructed for the expression of His-tagged MgrA or WalR. Overnight culture of these bacterial cells were diluted to an $OD_{600}$ of 0.05 in 100 mL TSB with 15 µg/mL chloramphenicol and grown to an $OD_{600}$ of 1.0 at 37 °C. One milliliter 0.5 M L-arginine were added as the substrate of NOS to induce the production of NO for 1 h. Cell pellets with 200 $OD_{600}$ were harvested by centrifugation at 5000 × *g* for 10 min and resuspended in 6.5 mL lysis buffer (50 mM Tris pH 8.0, 1% Triton X-100, 0.5 mM DTT) with 100 µg/mL lysostaphin (LSPN-50, AMBI) for 30 min incubation at 37 °C with shaking at 220 rpm. Then proteins were purified with a His-tag Protein Purification Kit (P2226, Beyotime) following the manufacturer's recommendation. The protein concentrations were determined using a Bradford Protein Assay Kit with bovine serum albumin as the standard.

Protein samples were prepared at 0.85 mg/mL. For each 200 µL sample, 7.3 mL HENS buffer, 480 µL 25% SDS and 160 µL 2 M methyl methanethiosulfonate (MMTS) were added, vortexed vigorously for 1 min to mix and incubated for 90 min at 50 °C with vortexing every 5 min to block free cysteine thiols. Protein precipitation was performed by adding six volumes of acetone pre-chilled at −20 °C and freezing at −20 °C for at least 1 h to remove MMTS. Samples were centrifuged at 10,000 × *g* for 5 min at 4 °C, washed three times with 2 mL acetone and resuspended in 50 µL HENS buffer. To label S-nitrosylated proteins, samples were added with 1 µL labeling reagent and 2 µL of 1 M sodium L-ascorbate and incubated for 90 min at 30 °C with shaking. Protein was precipitated by adding six volumes of pre-chilled acetone and freezing at −20 °C for at least 1 h to stop labeling, and resuspended in 40 µL HENS buffer. S-nitrosylation of equal concentrations of protein were detected by western blot. S-nitrosylated protein was detected with an anti-TMT antibody (90075, Pierce) with a dilution of 1:1000. Total protein was detected with an anti-His antibody (M30975, BOSTER) with a dilution of 1:1000. Full scans of the blots have been supplied in the Source Data file.

### Chromatin immunoprecipitation assay

The chromatin immunoprecipitation assay was adapted and modified from a previous study[64]. *S. aureus* strains were grown in TSB at 37 °C to an $OD_{600}$ of 1. Cells were pelleted at 5000 × *g* for 3 min at 4 °C and cross-linked with 1% formaldehyde in PBS (pH 7.4) for 15 min at room temperature. The cross-link was stopped by adding glycine to a final concentration of 125 mM with incubation for 10 min at room temperature. Cells were then pelleted and washed twice with 10 mL ice-cold PBS. Five $OD_{600}$ of pellets were resuspended in 500 µL lysis buffer (50 mM Tris pH 8.0, 1% Triton X-100, 0.5 mM DTT) with 100 µg/mL lysostaphin and incubated at 37 °C for 30 min. The lysed cells were added with Complete Protease Inhibitor Cocktail (K1007, APExBIO) and placed on ice to cool. Samples were then sonicated with a Bioruptor Plus sonication device (Diagenode) at low power for seven cycles with 15 s on and 15 s pulses. After centrifugation at 12,000 × *g* for 15 min at 4 °C, the DNA concentration of supernatant was adjusted to 40 µg/mL, and then diluted 1:100 in ChIP buffer (16.7 mM Tris pH 8.0, 150 mM NaCl, 1% Triton X-100, 2 mM EDTA). Five microliters of the diluted samples were saved as input sample at −20 °C, and another 500 µL was incubated with 2 µg of anit-MgrA polyclonal antibody (commercially prepared by MerryBio) or anit-WalR polyclonal antibody (commercially prepared by Genscript) or the corresponding IgG antibody (2729S, Cell Signaling Technology) overnight at 4 °C with gentle rotation. Then, 20 µL Protein A-Dynabeads slurry (10001D, Invitrogen) was added and incubated at 4 °C for another 3 h. The beads were washed twice with 1 mL of ice-cold buffer sequentially: wash buffer 1 (20 mM Tris pH 8.0, 150 mM NaCl, 0.1% SDS, 1% Triton X-100, 2 mM EDTA), wash buffer 2 (20 mM Tris pH 8.0, 350 mM NaCl, 0.1% SDS, 1% Triton X-100, 2 mM EDTA), LiCl buffer (10 mM Tris pH 8.0, 0.25 M LiCl, 1% NP40, 1% sodium deoxycholate; 1 mM EDTA) and TE buffer (10 mM

Tris pH 8.0, 1 mM EDTA). Rotate for 5 min with each buffer and carefully remove the supernatant. Resuspend beads in 200 µL elution buffer, incubate at 37 °C for 1 h and harvest the supernatant. Equal elution buffer was also added to the input samples. All samples were added with 8 µL NaCl (5 M) and incubated at 65 °C overnight. Four microliters RNase A (10 mg/mL) were added and incubated at 37 °C for 1 h. Then 4 µL proteinase K (10 mg/mL), 4 µL EDTA (0.5 M, pH 8.0) and 8 µL Tris-HCl (1 M, pH 6.5) were added and incubated at 45 °C for 3 h. The total DNA were extracted with 300 µL of 25:24:1 phenol/chloroform/isoamyl alcohol. DNA pellet was dissolve in 200 µL distilled water for qPCR quantification.

### Electrophoretic mobility shift assay

A FAM-labeled 217-bp fragment representing position 5–221 bp upstream of the *sarV* start codon, or a 281-bp fragment representing position −102 to +221 bp upstream of the *lytN* start codon, were incubated at room temperature for 1 min with various amounts of purified MgrA or MgrAC12S protein in 10 µL of binding buffer (20 mM Hepes pH 7.6, 1 mM EDTA, 10 mM $(NH_4)_2SO_4$, 30 mM KCl, 1 mM dithiothreitol and 0.2% (w/v) Tween-20). The reaction mixtures were analyzed in an 4.0% nondenaturing polyacrylamide gel. The band shifts were detected by Amersham Typhoon NIR (GE) and analyzed by ImageQuant TL 8.1. Full scans of the gels have been supplied in the Source Data file.

### Stable isotope label of S-nitrosylated protein

*S. aureus* strains were collected at $OD_{600}$ of 1 and resuspended in lysis buffer (8 M urea, 1% Triton-100, 10 mM dithiothreitol, and 1% Protease Inhibitor Cocktail). Samples were lysed with 0.1-mmdiameter-silica beads in a FastPrep-24 automated system (MP Biomedicals) and sonicated three times on ice using a high intensity ultrasonic processor (Scientz). The remaining debris was removed by centrifugation at 12,000 × *g* at 4 °C for 10 min. Protein concentration was determined with BCA kit (Beyotime Biotechnology) after the supernatant collection. Protein was prepared at 1–2 mg/mL in at least 1 mL of HENS Buffer and adjusted equally. One milligram protein was used for each reaction. Each sample were incubated with 20 mM MMTS to block free cysteine thiols. S-nitrosylated cysteines thiols were then reduced to free thiols by 10 mM sodium ascorbate and irreversibly labeled with isobaric iodoTMT according to the manufacturer's protocol for iodoTMTsixplex Isobaric Mass Tag Labeling Kit (90103, Thermo Scientific). Excess tag is removed by acetone precipitation.

### MS

LC−MS/MS was applied to identify S-nitrosylated proteins. Samples labeled with isobaric iodoTMT were digested with trypsin and fractionated into fractions by high pH reverse-phase HPLC using Thermo Betasil C18 column (5 µm particles, 10 mm ID, 250 mm length). Briefly, peptides were first separated with a gradient of 8% to 32% acetonitrile (pH 9.0) over 60 min into 60 fractions. Then, the peptides were combined, frozen and lyophilized using a vacuum concentrator. To enrich S-nitrosylated peptides, tryptic peptides dissolved in IP buffer (100 mM NaCl, 1 mM EDTA, 50 mM Tris-HCl, 0.5% NP40, pH 8.0) were incubated with pre-washed Anti-TMT Antibody beads (90076, Thermo Scientific) at 4 °C overnight with gentle shaking. Then the beads were washed four times with IP buffer and twice with $H_2O$. The bound peptides were eluted from the beads with 0.1% trifluoroacetic acid. Finally, the eluted fractions were vacuum-dried and desalted with C18 Zip Tips (Millipore) according to the manufacturer's instructions.

For LC−MS/MS analyses, the tryptic peptides were dissolved in solvent A (0.1% formic acid, 2% acetonitrile in water) and directly loaded onto a home-made reversed-phase analytical column (25-cm length, 75/100 µm i.d.). Peptides were separated with a gradient from 7% to 22% solvent B (0.1% formic acid in 90% acetonitrile) over 26 min,

22% to 35% in 8 min, 35% to 80% in 3 min, and climbed to 80% in 3 min and then holded at 80% for the last 4 min, all at a constant flowrate of 450 nl/min on an EASY-nLC 1000 UPLC system (Thermo Scientific). The peptides were subjected to NSI source followed by tandem mass spectrometry (MS/MS) in Orbitrap Fusion™ (Thermo Scientific) coupled online to the UPLC. The electrospray voltage applied was 2.0 kV. The $m/z$ scan range was 350–1550 for full scan. Intact peptides were detected in the Orbitrap at a resolution of 60,000. Up to 20 most abundant precursors were then selected for further MS/MS analyses with 15 s dynamic exclusion. The Higher-energy collision dissociation (HCD) fragmentation was performed at a normalized collision energy (NCE) of 35% and the fragments were detected in the Orbitrap at a resolution of 15,000. Automatic gain control (AGC) target was set at 5E4, with an intensity threshold of 5000 ions/s and a maximum injection time of 200 ms. The resulting MS/MS data were processed using MaxQuant search engine (v.1.5.2.8). Tandem mass spectra were searched against the NCBI *S. aureus* strain XN108 database (2992 entries) concatenated with reverse decoy database. Trypsin/P was specified as cleavage enzyme allowing up to 2 missing cleavages. The mass tolerance for precursor ions was set as 20 ppm in First search and 5 ppm in Main search, and the mass tolerance for fragment ions was set as 0.02 Da. Carbamidomethyl on Cys was specified as fixed modification. Acetylation on protein N-terminal, oxidation on Met, deamidation (NQ) on Gln and Asn, and Iodo TMT-6plex var were specified as variable modifications. FDR was adjusted to <1% and minimum score for peptides was set >40.

## Structural illustration

The structure of the MgrA dimer was downloaded from PDB: 2BV6[30], and the structure of the receiver domain of WalR was downloaded from PDB: 3F6P[43]. All the protein structure graphics were generated with PyMOL Molecular Graphics System.

## Statistical analysis

Statistical analysis were performed using Excel (Microsoft 2021) or Prism 9 (GraphPad) software. The number of biological replicates and statistical methods are indicated in the figure legends. Statistical significance was calculated using the two-sided unpaired Student's $t$-test or two-way ANOVA. All error bars show standard deviations (SD) of the mean. Statistical significance was defined as $p$ values ≤ 0.05.

## Reporting summary

Further information on research design is available in the Nature Portfolio Reporting Summary linked to this article.

# Data availability

A reporting summary for this article is available as Supplementary Information file. The main data supporting the findings of this study are available within the article and its Supplementary Information files. Additional details on datasets and protocols that support the findings of this study will be made available by the corresponding author upon reasonable request. The raw data of the proteomics generated in this study have been deposited in iProX database under accession code PXD041143 and are publicly available. Source data are provided with this paper.

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

## Acknowledgements
We thank Professor Xiancai Rao (Third Military Medical University, China) for the kind gift of XN108 strain. This work was supported by a grant from the National Key Research and Development Program of

China to B.S. (grant number: 2021YFC2300300); grants from National Natural Science Foundation of China to B.S. (grant number: 32070132; 31870126); a grant from National Natural Science Foundation of China to D.Y. (grant number: 31900132); a grant from Fundamental Research Funds for the Central Universities to B.S. (grant number: YD9100002013) and a grant from Beijing Natural Science Foundation to D.Y. (grant number: 7232052).

## Author contributions

B.S., D.Y. and X.S. conceived and designed the study. X.S. performed most of the experiments. Y.S. and Y.H. were responsible for the construction of mutant strains and part of the MIC experiments. X.S., D.Y. and B.S. analyzed the data and wrote the manuscript. All authors discussed the results and revised the manuscript.

## Competing interests

The authors declare no competing interests.
