## [Peer Review File · Nature Communications]

Transcription tuned by S-nitrosylation underlies a mechanism for *Staphylococcus aureus* to circumvent vancomycin killingREVIEWER COMMENTS

Reviewer #1 (Remarks to the Author):

This paper reports an interesting link between increased resistance of *Staphylococcus aureus* to the front-line antibiotic, vancomycin, and nitrosylation of two transcription repressors. The experiments appear to be sound, with results from initial experiments reinforced by multiple additional approaches. The authors conclude that nitrosylation of MgrA and WalR both result in stronger repression of genes encoding enzymes that degrade cell wall polymers. This results in thickening of the cell wall and hence presents a greater barrier to the entry of vancomycin to its targets. Note that some of the changes in vancomycin resistance reported, although statistically significant, are quite small. Nevertheless, this paper reports an interesting mechanism exploited by these bacteria to generate increased tolerance to an antibiotic. Data presented in figures 4 and 5 are especially convincing.

Although none of the experiments seem to be flawed, the authors should be invited to address the following questions.

SNP is a surrogate for NO that does not reliably report effects of NO itself. The authors cannot use NO-saturated water, or even more indirect NO donors such as prolyl-nonoate, because all of the experiments reported were completed with aerated cultures. Their parent strain grows well anaerobically, either by mixed acid fermentation or by nitrate, nitrite or fumarate respiration. Have they tried to determine the effect of NO itself on their key mutants and the parent? Is the parent strain resistant to higher concentrations of vancomycin during anaerobic growth? Is this resistance dependent upon NO production by NOS? Are the strains (where relevant) more resistant during anaerobic growth with nitrate or nitrite than with fumarate or glucose alone? Does anaerobic growth with nitrate compensate for the nos mutation?

Have the authors constructed mgrA walR double mutant? If so, is there an additive response to loss of vancomycin resistance? I appreciate that a double mutant might grow very poorly.

Figure 1 shows an extensive list of proteins that are nitrosylated and a sub-set that are transcription factors. Do the authors have quantitative data on the extent of nitrosylation of any of these proteins?

It would be nice to see results of control experiments with mutants in which one or two of the transcription factors in which the site of nitrosylation does not affect DNA binding have been deleted. If the main hypothesis is correct, this should not affect vancomycin resistance.

The accuracy of the English is far from perfect, but the paper is intelligible. It would benefit from careful revision by someone whose first language is English.

Reviewer #2 (Remarks to the Author):

S-nitrosylation by nitric oxide (NO) is an important means of gene regulation and immune function in humans. It has been recognized for a while that bacteria, including *Staphylococcus aureus*, also have NO synthases, which have been shown, including in *S. aureus*, to contribute to a variety of phenotypes, including resistance to antibiotics. However, the underlying mechanisms remain incompletely understood.

Shu et al. investigated the effects of NO in *S. aureus* by analyzing protein nitrosylation at cysteines by MS. They found that ~ 10% of proteins encoded in the *S. aureus* genome showed S-nitrosylation. They then focused on one regulatory protein that was among those hits, MgrA. They constructed a mutant in the nitric oxide synthase (nos) genes and a mutant in which the cysteine in MgrA found to be S-nitrosylated was exchanged for serine. Among the affected phenotypes they focused on vancomycin resistance and vancomycin resistance-related phenotypes such as cell wall thickness. They also investigated expression of known MgrA-regulated target genes and binding of MgrA and mutated MgrA to its promoter. Finally, they repeated a limited set of similar experiments with another hit, the regulator WalR.

The impact of bacterial nos on antibiotic resistance and the impact of MgrA and similar regulators on vancomycin resistance in *S. aureus* have been described. The significance of the present study

is thus closely linked to the demonstration that nitrosylation of regulatory proteins, with MgrA as example, is a key factor in the regulation of intermediate vancomycin resistance via cell wall thickening.

This would indeed be interesting, especially as it could serve as an example by which sort of mechanism S-nitrosylation and nos in *S. aureus* may have a general key impact on the control of *S. aureus* physiology. Unfortunately, the physiological role of nos regulation remains somehow unclear in the study as it is not shown under which physiological conditions it is expressed.

The main shortcoming of the study, however, is that it contains one major flaw in experimental setup. Namely, the authors analyze effects in an nos mutant and in the mgrA mutant with the amino acid position they showed to become nitrosylated (C12S). They never show that it is indeed nitrosylation at this amino acid position in MgrA that causes the observed effects. These effects could easily be independent, for example, C12 could be important for MgrA function independently of nitrosylation (cysteines are usually important for structure and function) and nitric oxide synthase could independently affect phenotypes (such as, just for example, by a direct impact on vancomycin as described for several agents by Gusarov et al. in their Science paper – ref. 20, or by other regulatory effects). While effects observed in the nos and C12S mutants are admittedly often comparable, this evidence remains indirect and inconclusive. Thus, for example, statements about the impact of S-nitrosylation when only the C12S mutant is compared to the WT are unfounded.

In the experiment in Fig. 4f, the authors come close to providing more conclusive evidence by using a nitrosylating agent (SNP) in the WT and C12S mutant background. However, effects are very small and the differences between significance and non-significance they take as evidence are virtually absent looking at the whole data. Furthermore, the t test they employed for statistical analysis is inappropriate for the analysis of more than 2 groups, as in these data.

One way to make the data more conclusive would be to include important controls, most notably analyzing effects of the C12A mutation in MgrA in a nos background and the impact of nitrosylating agents in an C12A mutant background as compared to WT (similar to what the authors did in Fig 4f, but in more experiments, and with appropriate statistical evaluation.)

Further points:

- I missed in the manuscript the formal evidence by protein MS analysis showing that in the absence of NOS (in the nos mutant), MgrA and other proteins do not become S-nitrosylated in *S. aureus*.
- Can the authors, based on their MS data, estimate how much of MgrA, or any given protein, becomes nitrosylated (at a given position)? Is MgrA a protein that shows a high level of nitrosylation as compared to other proteins?
- Several manuscripts that have reported on physiological phenotypes affected by the nos gene in *S. aureus* are not cited or discussed. How does the impact on MgrA and vanco resistance compare to these other reported effects? Is this an important or just a side effect of nos regulation?

Reviewer #3 (Remarks to the Author):

This manuscript is clear and concise and makes it easy for the reader to follow what was done and why (though this could be improved – see below). The effects seen represent relatively small changes in susceptibility, but this is potentially clinically relevant since the VISA phenotype typically occurs via sequential small increases in minimum inhibitory concentration (MIC).

Several approaches are used to test the hypothesis that S-nitrosylation contributes to the VISA phenotype. Some are relatively robust, but the biochemical assays are less convincing. However, the manuscript as a whole provides some evidence to support the interesting hypothesis and if this could be strengthened then the work would make a significant contribution to the field, as well as a wider contribution to the role of post-translational modifications in bacterial physiology.

In summary, to really justify the conclusions I think additional work is needed and clarification provided for several points:

I think it important to show that this mechanism is conserved across *S. aureus*, rather than in just this one strain. Preferably, this would involve additional delta-NOS mutants in other strains, but this could also be done biochemically by showing that NOS reduces vancomycin susceptibility/L-NAME increases susceptibility, and in parallel showing that the S-nitrosylated residues studied here are conserved in other strains.

Fig. 1. What proportions of proteins are S-nitrosylated? Do you just detect peptides with this modification or can you estimate what fraction of all the copies of MgrA in the cell are modified?

Fig. 2c. Can you provide evidence that growth inhibition with L-NAME is not due to L-NAME itself?

Fig. 4 & 6. The autolysis experiments are relatively convincing but the conclusion that this is due to altered autolysin production would be greatly strengthened by showing zymograms of cell wall extracts (e.g. see Bose et al., 2012 PLoS ONE PMID: 22860095). This is important because decreased autolysis could be due to e.g. altered teichoic acid abundance (e.g. Tiwari et al., 2018 AAC PMID:29735561).

Fig. 5. Are the recombinant proteins used in promoter binding assays S-nitrosylated? If not, then this calls into question whether this modification is needed and would suggest that the MgrAC12S mutation affects protein conformation or other functions independently of the modification.

I'm surprised that the authors did not make a walR/mgrA double mutant to assess the effect of S-nitrosylation on these 2 key regulators. If the aim is to understand the overall impact of NOS on vancomycin susceptibility then this experiment is needed.

The discussion was very limited, which is a shame because it would have been interesting to explore whether NOS could affect susceptibility to other antibiotics (e.g. VISA often have reduced susceptibility to daptomycin) and whether NOS would be a viable therapeutic target to block vancomycin resistance. It would also be useful to understand whether host NO produced by immune cells might affect antibiotic susceptibility.

Other points:

Line 70. Is NOS produced in all growth phases?

Line81. Please define HTH and explain the typical function of these domains.

Line 82. I think you need to lose the opening 'In' from this sentence.

Line 88. It would be useful here to explain why you looked at vancomycin susceptibility.

Line 96-97. I disagree that S-nitrosylation contributes to resistance development since this residue is also present in a vancomycin susceptible strain (8325). I think a more appropriate conclusion is that S-nitrosylation modulates vancomycin susceptibility. On a related note, it would be useful to know whether the C12 residue is conserved across *S. aureus*.

Line 127. This is growth inhibition, not killing that is being assayed.

Line 129. It would be useful to say here that VISA typically have thickened cell walls and explain how this is proposed to confer decreased susceptibility.

Line 137. It is not reasonable to conclude that decreased vancomycin susceptibility is due to a thickened cell wall in the WT relative to mutants. Whilst likely, this is not specifically tested.

Our point-by-point response to the reviewers' comments

Reviewer #1 (Remarks to the Author):

This paper reports an interesting link between increased resistance of *Staphylococcus aureus* to the front-line antibiotic, vancomycin, and nitrosylation of two transcription repressors. The experiments appear to be sound, with results from initial experiments reinforced by multiple additional approaches. The authors conclude that nitrosylation of MgrA and WalR both result in stronger repression of genes encoding enzymes that degrade cell wall polymers. This results in thickening of the cell wall and hence presents a greater barrier to the entry of vancomycin to its targets. Note that some of the changes in vancomycin resistance reported, although statistically significant, are quite small. Nevertheless, this paper reports an interesting mechanism exploited by these bacteria to generate increased tolerance to an antibiotic. Data presented in figures 4 and 5 are especially convincing.

Although none of the experiments seem to be flawed, the authors should be invited to address the following questions.

We greatly appreciate the reviewer's recognition of the strengths of our study and the value of the mechanism reported in this work. Below please find our detailed responses and descriptions of the additional experiments that were performed to address the reviewer's concerns.

SNP is a surrogate for NO that does not reliably report effects of NO itself. The authors cannot use NO-saturated water, or even more indirect NO donors such as prolyl-nonoate, because all of the experiments reported were completed with aerated cultures. Their parent strain grows well anaerobically, either by mixed acid fermentation or by nitrate, nitrite or fumarate respiration. Have they tried to determine the effect of NO itself on their key mutants and the parent? Is the parent strain resistant to higher concentrations of vancomycin during anaerobic growth? Is this resistance dependent upon NO production by NOS? Are the strains (where relevant) more resistant during anaerobic growth with nitrate or nitrite than with fumarate or glucose alone? Does anaerobic growth with nitrate compensate for the nos mutation?

We again thank the reviewer for the astute questions and constructive suggestion! As recommended, we performed vancomycin susceptibility experiments under anaerobic conditions (see below). However, we were unable to examine the effects of NO itself on the bacteria through direct injection of NO into the anaerobic incubator, since the anaerobic incubation system belonged to another lab whose ongoing experiments required N₂ injection, which would be disrupted by NO. However, we completely agree about the value of investigating the effects of the NO molecule itself on bacterial resistance. In future follow-up work, we plan to acquire a dedicated anaerobic incubator for our own use and continue exploration of the role of NO in *S. aureus* vancomycin

resistance.

To investigate whether the parent strain is resistant to higher concentrations of vancomycin during anaerobic growth, the WT strain was cultured in medium containing different concentrations of vancomycin under aerobic or anaerobic conditions. Then growth of the WT strain was measured during a 24 h period of aerobic growth or 48 h of anaerobic growth (since growth is markedly slower under anaerobic conditions) in MH broth containing a range of vancomycin concentrations. We found that the WT strain showed impaired growth at 8 $\mu\text{g/ml}$ vancomycin in aerobic cultures, whereas WT proliferation was attenuated at 5 $\mu\text{g/ml}$ vancomycin in anaerobic cultures (Fig. R1). To further verify the above results, we also compared differences in vancomycin resistance by aerobic or anaerobic WT cultures on agar plates. In agreement with the results of liquid cultures, we observed that vancomycin resistance was indeed lower in anaerobic WT cultures on solid medium. Specifically, WT growth was significantly inhibited on plates containing 4 $\mu\text{g/ml}$ vancomycin under anaerobic conditions, but could grow well on plates containing 8 $\mu\text{g/ml}$ vancomycin in aerobic conditions (Fig. R2). These collective results indicated that, in WT parental strain, vancomycin susceptibility was greater under anaerobic conditions than that in aerobic conditions. We are inclined to speculate that this effect is likely due to inhibition of NO production in the WT parental strain during anaerobic growth, as this process requires the incorporation of oxygen¹ and NO generation by NOS appears to be very important for vancomycin resistance in *S. aureus*.

Figure R1. Vancomycin MICs of the WT and Δnos strains cultured under aerobic or anaerobic conditions.

Figure R2. Plate-sensitivity assay of the WT and Δnos strains cultured under aerobic or anaerobic conditions.

To answer the question as to whether resistance during anaerobic growth is dependent on NO production by NOS, we also compared vancomycin resistance between WT and the Δnos strain during anaerobic growth. In medium containing vancomycin, the Δnos mutant exhibited similar growth under aerobic and anaerobic conditions (Fig. R1). In addition, Δnos mutant growth was inhibited on anaerobic plates containing 3 $\mu\text{g/ml}$ vancomycin, while anaerobic WT growth was inhibited on plates containing 4 $\mu\text{g/ml}$ vancomycin (Fig. R2), which is a smaller difference in resistance than that observed during aerobic growth. This small decrease in MIC may be attributable to the slightly slower growth of the Δnos mutant strain compared to the WT strain in the absence of vancomycin (Fig. R3). These results suggest that the regulation of vancomycin resistance is largely independent of NOS during anaerobic growth, which is consistent with the fact that NO production by NOS requires oxygen¹. These results collectively support that vancomycin resistance in *S. aureus* is predominantly controlled by NOS during aerobic growth, but largely independent of NOS activity during anaerobic growth.

Figure R3. Growth curve of the WT and Δnos strains.

We thank the reviewer for the excellent question regarding whether the bacteria showed higher resistance to vancomycin during anaerobic growth supplemented with nitrate or

nitrite than with fumarate or glucose. To test this possibility, we cultured WT or Δnos cells in complete synthetic medium PN² supplemented with nitrate, nitrite or fumarate, or glucose alone, which revealed that vancomycin resistance in anaerobic cultures was unaffected by supplementation with nitrate or nitrite rather than fumarate or glucose (Fig. R4). In *E. coli*, the addition of nitrate to anaerobic cultures can reportedly facilitate S-nitrosylation, in turn controlling transcription by OxyR³. However, no homolog of NOS has been identified in the *E. coli* genome⁴. Although *S. aureus* is a facultative anaerobe that can grow well in both aerobic and anaerobic conditions, *S. aureus* expresses NOS to modulate NO regulatory signals during aerobic growth. In our study, we found that approximately 10% of proteins in *S. aureus* undergo S-nitrosylation during aerobic growth, suggesting that S-nitrosylation is a prominent regulatory feature of aerobic growth in *S. aureus*. However, the efficiency of NO generation in medium supplemented with nitrate or nitrite during *S. aureus* growth under anaerobic conditions remains unknown, as does the proportion of S-nitrosylated proteins in these cells. Moreover, it is also unclear whether *S. aureus* uses S-nitrosylation as a regulatory signal for vancomycin resistance under anaerobic conditions. Thus, a better understanding of the effects of NO-mediated S-nitrosylation on antibiotic resistance under anaerobic conditions can help to resolve many questions regarding the mechanistic control of vancomycin resistance, which will be investigated in future and ongoing studies.

Figure R4. Vancomycin MICs of the WT and Δnos strains when cultured in PN medium supplemented with nitrate, nitrite or fumarate, or glucose alone under aerobic or anaerobic conditions.

Finally, in response to the reviewer's question as to whether anaerobic growth with nitrate can compensate for the *nos* mutation, we investigated the role of NOS on vancomycin resistance during anaerobic growth, and found that resistance was largely independent of NOS under anaerobic conditions, as shown above (Fig. R1 & 2). This may be attributable to the inhibition of NO production by NOS under anaerobic conditions, since this process requires the incorporation of oxygen¹. In addition,

supplementation with nitrate showed no obvious effects on vancomycin resistance in either WT or the Δnos mutant strain (Fig. R4). These data therefore indicate that nitrate supplementation during anaerobic growth does not compensate for the *nos* mutation.

Have the authors constructed *mgrA walR* double mutant? If so, is there an additive response to loss of vancomycin resistance? I appreciate that a double mutant might grow very poorly.

As advised by the reviewer, we constructed an *mgrAC12S/walRC67S* double mutant. We found that this double mutant grew as well as the WT strain, which probably because that either *mgrA* or *walR* single mutation did not cause growth defective. Rather, these mutants showed increased growth as compared to the parent strain (Fig. R5a, also see Supplementary Fig. 8a). The MIC of vancomycin was compared between the WT, *mgrAC12S*, *walRC67S*, and *mgrAC12S/walRC67S* strains. We found that the *mgrAC12S/walRC67S* mutant showed growth defects at 4 $\mu\text{g/ml}$ vancomycin, with no significant difference compared to the *mgrAC12S* or *walRC67S* mutant (Fig. R5b, also see Supplementary Fig. 8b), which is consistent with the observation that the Δnos strain retained a MIC of 4 $\mu\text{g/ml}$. This may be because although MgrA and WalR are both transcriptional regulators regulating similar genes involved in autolysin production, they are likely not collaborators, and thus do not have coordinated function and are not dependent on one another to regulate vancomycin resistance^{5,6}.

Besides, several genes, such as *vraTSR*, *graSR* and *rpoB*, were also reported important for VISA development⁷, likely independent of NO-mediated S-nitrosylation. Thus, the function of these genes may also contribute to vancomycin resistance in *S. aureus*. We have added these results in Supplementary Figure 8 and revised the Results section (See Line 298-314) in the manuscript.

Figure R5 (Supplementary Figure 8). Growth curve and vancomycin MICs of the WT and mutant strains.

Figure 1 shows an extensive list of proteins that are nitrosylated and a sub-set that are transcription factors. Do the authors have quantitative data on the extent of nitrosylation of any of these proteins?

Thanks for this insightful question. It would be both useful and interesting to know the

extent of the proteins are S-nitrosylated. In this study, to identify the endogenous S-nitrosylation sites, S-nitrosylated proteins were irreversibly labeled with isobaric iodoTMT reagent using a Labeling Kit (Details were shown in the Methods). This process enriches the S-nitrosylated target proteins before mass spectrometry (MS) analysis, since these modified proteins are typically present in very low concentrations in cells. Enrichment of the modified peptides is also the conventional strategy for studying post-translational modification, which is largely due to the characteristics of low level and instability of these modifications^{8,9}. Because only the modified peptides can be detected, it is hard to calculate the modified proportion of target protein, since there is no quantitative MS data for the remaining unmodified proportion of target protein. Conversely, quantification of the total amount of a protein through routine MS without specific enrichment for S-nitrosylated peptides will have a low probability of detecting modified sites. In addition, even if the unmodified population was separated, there is not only S-nitrosylation on Cys12 of MgrA; for example, phosphorylation and oxidation have been identified, other modifications may also exist. It is therefore difficult to calculate the proportion of an endogenously expressed protein that becomes S-nitrosylated, which has been identified as a challenging issue for proteomics.

It would be nice to see results of control experiments with mutants in which one or two of the transcription factors in which the site of nitrosylation does not affect DNA binding have been deleted. If the main hypothesis is correct, this should not affect vancomycin resistance.

We appreciate the reviewer's constructive suggestion. In this study, we demonstrated that the transcriptional regulation by NO-mediated S-nitrosylation is an important mechanism for control of vancomycin resistance in *S. aureus* using two well-known transcription factors, MgrA and WalR, which have been proposed to contribute to regulating antibiotic resistance, as examples. Our ChIP-qPCR results show that the NO signal affects downstream transcriptional regulation by modulating the ability of MgrA and WalR to bind DNA, thus providing evidence that the transcriptional regulatory effects of NO are mediated by protein S-nitrosylation.

Although DNA binding is a central property of transcriptional regulators, it is not the only mode of controlling gene transcription, and regulation of transcription factor activity is often complex, involving multiple inter- and/or intracellular signal transduction pathways. For example, transcription factors commonly contain sites for sensing stimuli or sequence motifs for interacting with other factors that affect its regulatory activity, albeit without directly altering its DNA binding. These sites may also function in the transcriptional activation of the regulator itself or its interaction partners. Thus, it is both difficult and complicated to exclude the function of a site in affecting DNA binding and also possible that S-nitrosylation sites affect the binding of other ligands that participate in transcriptional regulation without influencing DNA binding activity.

The accuracy of the English is far from perfect, but the paper is intelligible. It would

benefit from careful revision by someone whose first language is English.

We thank the reviewer's comment that the paper is intelligible. As advised, we've found a native speaker to help improving the language of the manuscript.

Reviewer #2 (Remarks to the Author):

S-nitrosylation by nitric oxide (NO) is an important means of gene regulation and immune function in humans. It has been recognized for a while that bacteria, including *Staphylococcus aureus*, also have NO synthases, which have been shown, including in *S. aureus*, to contribute to a variety of phenotypes, including resistance to antibiotics. However, the underlying mechanisms remain incompletely understood.

Shu et al. investigated the effects of NO in *S. aureus* by analyzing protein nitrosylation at cysteines by MS. They found that ~ 10% of proteins encoded in the *S. aureus* genome showed S-nitrosylation. They then focused on one regulatory protein that was among those hits, MgrA. They constructed a mutant in the nitric oxide synthase (*nos*) genes and a mutant in which the cysteine in MgrA found to be S-nitrosylated was exchanged for serine. Among the affected phenotypes they focused on vancomycin resistance and vancomycin resistance-related phenotypes such as cell wall thickness. They also investigated expression of known MgrA-regulated target genes and binding of MgrA and mutated MgrA to its promoter. Finally, they repeated a limited set of similar experiments with another hit, the regulator WalR.

The impact of bacterial *nos* on antibiotic resistance and the impact of MgrA and similar regulators on vancomycin resistance in *S. aureus* have been described. The significance of the present study is thus closely linked to the demonstration that nitrosylation of regulatory proteins, with MgrA as example, is a key factor in the regulation of intermediate vancomycin resistance via cell wall thickening.

This would indeed be interesting, especially as it could serve as an example by which sort of mechanism S-nitrosylation and *nos* in *S. aureus* may have a general key impact on the control of *S. aureus* physiology. Unfortunately, the physiological role of *nos* regulation remains somehow unclear in the study as it is not shown under which physiological conditions it is expressed.

We thank the reviewer for the supportive and constructive comments towards improving the quality of our study. The authors agree that this work expands our mechanistic understanding of the roles of S-nitrosylation and NOS in *S. aureus* physiology in general, in addition to their specific functions in vancomycin resistance. Below, please find our responses to each comment or question and descriptions of additional experiments that were performed to address their concerns.

To address the reviewer's concern, we have measured the transcription of *nos* during different growth phases in medium supplemented with or without vancomycin. The

qRT-PCR results showed that *nos* displayed a constant transcription during the whole growth phase, with no more than 2-fold induction at the lag phase. We also found that 2 $\mu\text{g/ml}$ vancomycin treatment did not significantly change *nos* expression, with a slightly decrease at the lag phase (Fig. R6), suggesting a constant expression of *nos* in *S. aureus*, which is in consistent with the previous study^{10,11}.

Figure R6. Transcription levels of *nos* in bacteria treated with or without vancomycin at different growth phase.

The main shortcoming of the study, however, is that it contains one major flaw in experimental setup. Namely, the authors analyze effects in an *nos* mutant and in the *mgrA* mutant with the amino acid position they showed to become nitrosylated (C12S). They never show that it is indeed nitrosylation at this amino acid position in MgrA that causes the observed effects. These effects could easily be independent, for example, Cys12 could be important for MgrA function independently of nitrosylation (cysteines are usually important for structure and function) and nitric oxide synthase could independently affect phenotypes (such as, just for example, by a direct impact on vancomycin as described for several agents by Gusarov et al. in their Science paper – ref. 20, or by other regulatory effects). While effects observed in the *nos* and C12S mutants are admittedly often comparable, this evidence remains indirect and inconclusive. Thus, for example, statements about the impact of S-nitrosylation when only the C12S mutant is compared to the WT are unfounded.

In the experiment in Fig. 4f, the authors come close to providing more conclusive evidence by using a nitrosylating agent (SNP) in the WT and C12S mutant background. However, effects are very small and the differences between significance and non-significance they take as evidence are virtually absent looking at the whole data. Furthermore, the t test they employed for statistical analysis is inappropriate for the analysis of more than 2 groups, as in these data.

One way to make the data more conclusive would be to include important controls, most notably analyzing effects of the C12A mutation in MgrA in a *nos* background and the impact of nitrosylating agents in an C12A mutant background as compared to WT (similar to what the authors did in Fig 4f, but in more experiments, and with appropriate statistical evaluation.)

We thank the reviewer for these astute questions and valuable suggestions! We agree with the reviewer that the effects of *nos* knockout could be independent of *mgrAC12S*

mutation, and vice versa. In the original version of the manuscript, we describe complementary experiments in which the Δnos mutant and the *mgrAC12S* strain show similar vancomycin resistance phenotypes, providing initial verification that the absence of MgrA S-nitrosylation or the absence of NO production both negatively impact resistance. If the two strains showed contradictory results, then the mechanism responsible for decreased vancomycin resistance would be attributable to another pathway other than NOS-dependent NO production. After verifying that the effects of these mutations were highly similar, we then used an NO donor to investigate how NO-mediated S-nitrosylation impacts vancomycin resistance in multiple strains (Figure 4). Although, as the reviewer noted, the differences in Figure 4f are small, they may be clinically relevant since the VISA phenotype typically emerges incrementally through small increases in MIC. The significance of the WT and each mutant was analyzed, as there are only two groups for each comparison analysis, Student's t-test was thus employed here. Furthermore, we also used ChIP-qPCR to evaluate the effects of S-nitrosylation on the capacity of MgrA to bind different gene promoters in WT, *mgrAC12S* or Δnos strain. This experiment thus excludes the effects of possible structural changes in MgrA caused by the Cys12 mutation. We feel these data provide strong evidence that NO-mediated S-nitrosylation in *S. aureus* indeed affects the ability of MgrA to bind target DNA and modulate vancomycin resistance.

We again thank reviewer's excellent suggestion to help improving the study. To provide additional evidence for the proposed regulatory mechanism, we constructed an *mgrAC12A* mutant in the WT and Δnos strains with allelic replacement. We first measured the vancomycin MIC of these strains. Consistent with results from the *mgrAC12S* mutant, the *mgrAC12A* mutant also showed a decreased MIC at 4 $\mu\text{g/ml}$ (Fig. R7; also see Supplementary Fig. 4) in comparison to the WT strain. In addition, the *mgrAC12A* mutation in the *nos* knockout background did not further decrease the vancomycin MIC (Fig. R7; also see Supplementary Fig. 4). Then, the effect of NO donor on the growth of the *mgrAC12A* mutant was measured in MH medium containing 1 $\mu\text{g/ml}$ vancomycin. We found that addition of 0, 1, 5, or 20 μM SNP (displayed as N-0, N-1, N-5 or N-20) could not compensate the defective growth of the *mgrAC12A* mutant and furthermore exhibited a significant, dose-dependent inhibitory effect (Fig. R8; also see Supplementary Fig. 5), which was consistent with our findings in the *mgrAC12S* strain.

The autolytic activity and transcription of genes that involved in autolysis in the *mgrAC12A* mutant and WT strains were also compared. We found that the *mgrAC12A* mutant showed increased autolytic activity comparable to that of the *mgrAC12S* mutant, which could not be compensated by treatment with 5 μM SNP (displayed as N-5) (Fig. R9a; also see Supplementary Fig. 6a). qRT-PCR analyses showed that *lytN* and *sarV* were upregulated in the *mgrAC12A* mutant compared with the WT strain (Fig. R9b; also see Supplementary Fig. 6b), which is also consistent with results in the *mgrAC12S* mutant strain. Furthermore, we treated the *mgrAC12A* mutant strains with 0.2 or 1 mM SNP as did in Figure 4f. NO donor treatment did not induce significant changes in *lytN*

and *sarV* expression in the *mgrAC12A* mutant strain (Fig. R9c; also see Supplementary Fig. 6c), which further confirmed our conclusion that removal of the S-nitrosylation from MgrA Cys12 severely impaired NO-mediated signal transduction to *lytN* and *sarV*. Collectively, these data demonstrate a role of S-nitrosylation at MgrA Cys12 in modulating vancomycin susceptibility in *S. aureus*. We have added these results in Supplementary Figure S4-6 and revised the Results section (See Line 245-269) in the manuscript.

Figure R7 (Supplementary Figure 4). Vancomycin MICs of the WT and other mutant strains.

Figure R8 (Supplementary Figure 5). Growth curve of the WT and *mgrAC12A* mutant strains in medium containing vancomycin when added with different concentrations of SNP.

Figure R9 (Supplementary Figure 6). The autolytic activity and the transcription levels of *lytN* and *sarV* in medium supplemented with or without SNP were measured

in WT and *mgrAC12A* strains.

Further points:

- I missed in the manuscript the formal evidence by protein MS analysis showing that in the absence of NOS (in the *nos* mutant), MgrA and other proteins do not become S-nitrosylated in *S. aureus*.

Different from conventional MS analysis, the detection of S-nitrosylated proteins requires irreversible labeling with isobaric iodoTMT reagent. Since very few companies offer this service, the process is considerably more expensive than conventional MS-based proteomics. Due to limited funding, we are currently unable to submit other samples for comparison of S-nitrosylated sites between the WT and mutant strains to supplement the results of S-nitrosylation site detection in the Δnos mutant. In addition to cost, the experimental strategy for comparing the extent of protein modification between two strains is considerably different from the strategy used for identification of modified sites in a single strain. Specifically, to strictly compare the intensity of modified protein signals between different strains, the WT and Δnos mutant strains are mixed together for sample labeling and loading. Only peptides with a positive signal in all samples are then collected for further analysis and comparison, and if the target protein in one strain shows no signal, then the overall output value of the ratio is zero, and no information about signal intensity is available for individual sample. Thus, considering this issue along with the experimental cost, we preferred to use direct identification of the S-nitrosylated sites as our approach.

We thank the reviewer for raising this excellent question. We agree that quantification of changes in S-nitrosylation at specific sites in the Δnos mutant, especially results showing that this modification is decreased or abolished, would directly resolve questions about the role of NOS in controlling S-nitrosylation. We have plans to pursue this line of experimentation in future work.

- Can the authors, based on their MS data, estimate how much of MgrA, or any given protein, becomes nitrosylated (at a given position)? Is MgrA a protein that shows a high level of nitrosylation as compared to other proteins?

Thanks for this insightful question. It would be both useful and interesting to know the extent of the proteins are S-nitrosylated. In this study, to identify the endogenous S-nitrosylation sites, S-nitrosylated proteins were irreversibly labeled with isobaric iodoTMT reagent using a Labeling Kit (Details were shown in the Methods). This process enriches the S-nitrosylated target proteins before mass spectrometry (MS) analysis, since these modified proteins are typically present in very low concentrations in cells. Enrichment of the modified peptides is also the conventional strategy for studying post-translational modification, which is largely due to the characteristics of low level and instability of these modifications^{8,9}. Because only the modified peptides

can be detected, it is hard to calculate the modified proportion of target protein, since there is no quantitative MS data for the remaining unmodified proportion of target protein. Conversely, quantification of the total amount of a protein through routine MS without specific enrichment for S-nitrosylated peptides will have a low probability of detecting modified sites. In addition, even if the unmodified population was separated, there is not only S-nitrosylation on Cys12 of MgrA; for example, phosphorylation and oxidation have been identified, other modifications may also exist. It is therefore difficult to calculate the proportion of an endogenously expressed protein that becomes S-nitrosylated, which has been identified as a challenging issue for proteomics.

- Several manuscripts that have reported on physiological phenotypes affected by the *nos* gene in *S. aureus* are not cited or discussed. How does the impact on MgrA and vanco resistance compare to these other reported effects? Is this an important or just a side effect of *nos* regulation?

We're sorry for missing these reports on physiological phenotypes affected by the *nos* gene in *S. aureus*, and we have included these papers in the manuscript and revised the text in the discussion part (See Line 337-336) as follows: NOS has also been reported to influence *S. aureus* virulence^{12,13}, sensitivity to oxidative stress^{10,13} and heme stress¹⁴; it also plays essential roles in regulating electron transfer to maintain membrane bioenergetics¹⁵ and modulates aerobic respiratory metabolism and cell physiology¹⁶. Bacterial NOS-derived NO also plays important roles in regulating the formation, swarming, and dispersal of bacterial biofilms^{17,18}. These findings suggest a key role of bacterial NOS or NOS-derived NO in the global regulation of *S. aureus* physiological phenotypes. It is thus worth exploring the mechanisms by which NOS is regulated. The regulatory mechanism mediated by S-nitrosylation may be universal and could be applied to regulation of other physiological phenotypes, and it should therefore be further explored in the future.

To answer the question, how does the impact of MgrA and vancomycin resistance compare with effects described in other reports; is this an important or just a side effect of *nos* regulation: MgrA is a global regulatory protein in *S. aureus* involved in autolytic activity, multidrug resistance, and virulence^{19,20}. However, its underlying mechanisms remain elusive. S-nitrosylation of the *S. aureus* transcription factor AgrA by host NO was found to inhibit *S. aureus* virulence by influencing the ability of AgrA to bind DNA⁸. The function of S-nitrosylation generated from endogenous NO in *S. aureus* is still unclear. In this study, mass spectrometry data provide solid evidences that endogenous MgrA is S-nitrosylated in a vancomycin-intermediate *S. aureus* (VISA) strain. Focusing on this modification, we investigated the regulatory mechanism of vancomycin resistance mediated by the S-nitrosylation of MgrA in *S. aureus*, and demonstrated that transcriptional regulation via MgrA S-nitrosylation underlies a mechanism for NO-mediated *S. aureus* antibiotic resistance. Moreover, this S-nitrosylation-mediated mechanism of vancomycin resistance was also observed in another transcriptional regulator, WalR. These collective results support our findings that S-nitrosylation of MgrA and regulation of vancomycin resistance are not side effects of NOS, but rather important bacterial mechanisms for overcoming antibiotic

cytotoxicity that may be generalized to other bacteria. These findings expand the scope of known mechanisms leading to antibiotic resistance, and broadly contribute to our understanding of the role of post-translational modifications in bacterial physiology.

Reviewer #3 (Remarks to the Author):

This manuscript is clear and concise and makes it easy for the reader to follow what was done and why (though this could be improved – see below). The effects seen represent relatively small changes in susceptibility, but this is potentially clinically relevant since the VISA phenotype typically occurs via sequential small increases in minimum inhibitory concentration (MIC).

Several approaches are used to test the hypothesis that S-nitrosylation contributes to the VISA phenotype. Some are relatively robust, but the biochemical assays are less convincing. However, the manuscript as a whole provides some evidence to support the interesting hypothesis and if this could be strengthened then the work would make a significant contribution to the field, as well as a wider contribution to the role of post-translational modifications in bacterial physiology.

In summary, to really justify the conclusions I think additional work is needed and clarification provided for several points:

We thank the reviewer for their positive comments regarding our manuscript and the significance of this study to the field of bacterial physiology. Below, please find our point-by-point response and descriptions of additional experiments conducted to address the reviewer's concerns.

I think it important to show that this mechanism is conserved across *S. aureus*, rather than in just this one strain. Preferably, this would involve additional delta-NOS mutants in other strains, but this could also be done biochemically by showing that NOS reduces vancomycin susceptibility/L-NAME increases susceptibility, and in parallel showing that the S-nitrosylated residues studied here are conserved in other strains.

We appreciate the reviewer's constructive suggestion. As advised, we treated another VISA strain Mu50 with L-NAME, and found that the growth of Mu50 in the medium containing 2 µg/ml vancomycin (displayed as V-2) was further inhibited by addition of 40 mM L-NAME (displayed as V-2 L-40) (Fig. R10; also see Fig. 2d in the manuscript), which is consistent with the result we observed in XN108. We have added the results in Figure 2d and revised the Results section in the manuscript (See Line 118-130). In addition, two previously constructed vancomycin-sensitive *S. aureus* strains, NCTC8325 and MW2, were used to compare the vancomycin susceptibility after NOS deletion. We found that *nos* knockout caused an increase in vancomycin susceptibility (Fig. R11). Furthermore, *nos* deletion in *S. aureus* strains USA300 has also been reported to cause a decrease in vancomycin MIC (Fig. R12)¹³. Together, these data

suggested that NOS-mediated S-nitrosylation in modulating vancomycin susceptibility could be a conserved mechanism across *S. aureus*.

Figure R10 (Figure 2d). Growth curve of Mu50 in normal MH medium, medium containing vancomycin, or medium containing vancomycin supplemented with L-NAME.

Figure R11. Vancomycin MICs of the WT and Δnos mutant strains derived from NCTC8325 or MW2.

MRSA Strain	Vanco MIC ($\mu\text{g/ml}$)
WT	1.0
ΔNOS	0.5
$\Delta\text{NOS} + \text{pNOS}$	1.0

Figure R12. Vancomycin MICs of the USA300 WT and Δnos mutant strains¹³.

To analyze the conservation of the S-nitrosylated residue in *S. aureus*, the amino acid was aligned among different *S. aureus* strains that have conserved MgrA protein by using Clustal Omega. The alignment results showed that the Cys12 in MgrA (Fig. R13) or C67 in WalR (Fig. R14) was conserved among these strains.

MgrA ★ C12

```

XN108 1 MSDQHNLKEQLCFSLYNAQRQVNRYYSNKVFKKYNLTYPQFLVLT I LWD 49
Mu50 1 MSDQHNLKEQLCFSLYNAQRQVNRYYSNKVFKKYNLTYPQFLVLT I LWD 49
MW2 1 MSDQHNLKEQLCFSLYNAQRQVNRYYSNKVFKKYNLTYPQFLVLT I LWD 49
N315 1 MSDQHNLKEQLCFSLYNAQRQVNRYYSNKVFKKYNLTYPQFLVLT I LWD 49
NCTC8325 1 MSDQHNLKEQLCFSLYNAQRQVNRYYSNKVFKKYNLTYPQFLVLT I LWD 49

XN108 50 ESPVNVKKVVTELALDTGTVSPLLKRMEQVDL I KRERSEVDQREVF I HL 98
Mu50 50 ESPVNVKKVVTELALDTGTVSPLLKRMEQVDL I KRERSEVDQREVF I HL 98
MW2 50 ESPVNVKKVVTELALDTGTVSPLLKRMEQVDL I KRERSEVDQREVF I HL 98
N315 50 ESPVNVKKVVTELALDTGTVSPLLKRMEQVDL I KRERSEVDQREVF I HL 98
NCTC8325 50 ESPVNVKKVVTELALDTGTVSPLLKRMEQVDL I KRERSEVDQREVF I HL 98

XN108 99 TDKSET I RPELSNASDKVASASSLSQDEVKELNRL LGKV I HAFDETKEK 147
Mu50 99 TDKSET I RPELSNASDKVASASSLSQDEVKELNRL LGKV I HAFDETKEK 147
MW2 99 TDKSET I RPELSNASDKVASASSLSQDEVKELNRL LGKV I HAFDETKEK 147
N315 99 TDKSET I RPELSNASDKVASASSLSQDEVKELNRL LGKV I HAFDETKEK 147
NCTC8325 99 TDKSET I RPELSNASDKVASASSLSQDEVKELNRL LGKV I HAFDETKEK 147

```

Figure R13. The alignment of MgrA amino acid among different *S. aureus* strains.

WalR ★ C67

```

XN108 1 MQMARKVVVVVDEKPIADILEFNLKKEGYDVYCA YDGNDAVDL I YEEEPD I VLLD I MLP 59
Mu50 1 MQMARKVVVVVDEKPIADILEFNLKKEGYDVYCA YDGNDAVDL I YEEEPD I VLLD I MLP 59
MW2 1 - - MARKVVVVVDEKPIADILEFNLKKEGYDVYCA YDGNDAVDL I YEEEPD I VLLD I MLP 57
N315 1 - - MARKVVVVVDEKPIADILEFNLKKEGYDVYCA YDGNDAVDL I YEEEPD I VLLD I MLP 57
NCTC8325 1 - - MARKVVVVVDEKPIADILEFNLKKEGYDVYCA YDGNDAVDL I YEEEPD I VLLD I MLP 57

XN108 60 GRDGMVCREVRKKYEMPI I MLTAKDSE I DKVLG LELGADDYVTKPFSTREL I ARVKAN 118
Mu50 60 GRDGMVCREVRKKYEMPI I MLTAKDSE I DKVLG LELGADDYVTKPFSTREL I ARVKAN 118
MW2 58 GRDGMVCREVRKKYEMPI I MLTAKDSE I DKVLG LELGADDYVTKPFSTREL I ARVKAN 116
N315 58 GRDGMVCREVRKKYEMPI I MLTAKDSE I DKVLG LELGADDYVTKPFSTREL I ARVKAN 116
NCTC8325 58 GRDGMVCREVRKKYEMPI I MLTAKDSE I DKVLG LELGADDYVTKPFSTREL I ARVKAN 116

XN108 119 LRRHYSQPAQDTGNVTNE I T I KD I V I YPDAYS I KKRGED I ELTHREFELFHYLSKHMGG 177
Mu50 119 LRRHYSQPAQDTGNVTNE I T I KD I V I YPDAYS I KKRGED I ELTHREFELFHYLSKHMGG 177
MW2 117 LRRHYSQPAQDTGNVTNE I T I KD I V I YPDAYS I KKRGED I ELTHREFELFHYLSKHMGG 175
N315 117 LRRHYSQPAQDTGNVTNE I T I KD I V I YPDAYS I KKRGED I ELTHREFELFHYLSKHMGG 175
NCTC8325 117 LRRHYSQPAQDTGNVTNE I T I KD I V I YPDAYS I KKRGED I ELTHREFELFHYLSKHMGG 175

XN108 178 VMTREHLLQTVWGYDYFGDVRTVDVT I RRLREK I EDDPSHPEY I VTRRGVGYFLQQHE 235
Mu50 178 VMTREHLLQTVWGYDYFGDVRTVDVT I RRLREK I EDDPSHPEY I VTRRGVGYFLQQHE 235
MW2 176 VMTREHLLQTVWGYDYFGDVRTVDVT I RRLREK I EDDPSHPEY I VTRRGVGYFLQQHE 233
N315 176 VMTREHLLQTVWGYDYFGDVRTVDVT I RRLREK I EDDPSHPEY I VTRRGVGYFLQQHE 233
NCTC8325 176 VMTREHLLQTVWGYDYFGDVRTVDVT I RRLREK I EDDPSHPEY I VTRRGVGYFLQQHE 233

```

Figure R14. The alignment of WalR amino acid among different *S. aureus* strains.

Fig. 1. What proportions of proteins are S-nitrosylated? Do you just detect peptides with this modification or can you estimate what fraction of all the copies of MgrA in the cell are modified?

Thanks for this insightful question. It would be both useful and interesting to know the extent of the proteins are S-nitrosylated. In this study, to identify the endogenous S-nitrosylation sites, S-nitrosylated proteins were irreversibly labeled with isobaric iodoTMT reagent using a Labeling Kit (Details were shown in the Methods). This process enriches the S-nitrosylated target proteins before mass spectrometry (MS) analysis, since these modified proteins are typically present in very low concentrations in cells. Enrichment of the modified peptides is also the conventional strategy for studying post-translational modification, which is largely due to the characteristics of low level and instability of these modifications^{8,9}. Because only the modified peptides can be detected, it is hard to calculate the modified proportion of target protein, since there is no quantitative MS data for the remaining unmodified proportion of target protein. Conversely, quantification of the total amount of a protein through routine MS without specific enrichment for S-nitrosylated peptides will have a low probability of detecting modified sites. In addition, even if the unmodified population was separated,

there is not only S-nitrosylation on Cys12 of MgrA; for example, phosphorylation and oxidation have been identified, other modifications may also exist. It is therefore difficult to calculate the proportion of an endogenously expressed protein that becomes S-nitrosylated, which has been identified as a challenging issue for proteomics.

Fig. 2c. Can you provide evidence that growth inhibition with L-NAME is not due to L-NAME itself?

We thank the reviewer for the excellent question. L-NAME has been applied as a specific inhibitor for NOS activity^{10,15}. We hypothesized that if the growth inhibition observed in Figure 2c was due to nonspecific activity of L-NAME, treatment of the Δnos strain with would also induce significant growth inhibition. However, addition of 5, 10, or 40 μM L-NAME (displayed as L-5, L-10 or L-40) did not inhibit Δnos growth. In fact, it showed a small growth stimulation effect, whereas a growth inhibition was found in the WT when added with 40 mM L-NAME (Fig. R15a and 15b; also see Supplementary Fig. 2a and 2b). This is possibly due to the fact that L-NAME is an L-Arginine analog and bacteria may use it for growth when NOS is absent in cells. Thus, the results suggested that the growth inhibition caused by L-NAME was a result of NOS activity inhibition rather than an effect nonspecific to L-NAME. We have added these results in Supplementary Figure 2 and revised the Result section and figure legends in the manuscript and supplemental (See Line 120-130).

Figure R15 (Supplementary Figure 2). The effect of NOS inhibitor L-NAME on the growth of WT or Δnos strain.

Fig. 4 & 6. The autolysis experiments are relatively convincing but the conclusion that this is due to altered autolysin production would be greatly strengthened by showing zymograms of cell wall extracts (e.g. see Bose et al., 2012 PLoS ONE PMID: 22860095). This is important because decreased autolysis could be due to e.g., altered teichoic acid abundance (e.g., Tiwari et al., 2018 AAC PMID:29735561).

We thank the reviewer for the good suggestion. We performed zymograms of cell wall extracts as previously described^{21,22}. Total cell wall extract from WT and mutant strains were separated by SDS-PAGE on 12% acrylamide gels containing 2.5×10^9 CFU ml^{-1} of heat-killed mid-exponential phase *S. aureus* as a substrate. Gels were washed and

incubated in reaction buffer overnight at 37 °C, then stained with methylene blue. Because the autolysins can lyse *S. aureus* cells, it will thus appear as discrete white bands on the gel if there are autolysins in the specific position on the gel. We found that, compared to the WT strain, three bands in the Δnos , five bands in the *mgrAC12S*, and two bands in the *walRC67S* mutant (Fig. R16a-c) noted with black arrows showed increased levels. Despite the exact autolysin for each band is unclear, the expression upregulation of these autolysins is consistent with the increased autolytic activity and elevated transcriptional level of genes involved in autolysis we found in Figure 4&6, suggesting that the increased autolysis activity of these mutants is largely due to the increased autolysin production.

Figure R16. zymography bacteriolytic profiles for the cell wall extracts.

Fig. 5. Are the recombinant proteins used in promoter binding assays S-nitrosylated? If not, then this calls into question whether this modification is needed and would suggest that the *MgrAC12S* mutation affects protein conformation or other functions independently of the modification.

We thank the reviewer for this excellent question. To address this issue, we detected the S-nitrosylation signal of His-tagged *MgrA* expressed in *E. coli* using an S-Nitrosylation Western Blot Kit (90105, Pierce). In this assay, the unmodified cysteines are first blocked with the sulfhydryl-reactive compound Methyl Methanethiosulfonate (MMTS). S-nitrosylated cysteines are then selectively reduced with sodium ascorbate in HENS buffer to specifically and irreversibly label S-nitrosylated cysteine thiols with iodoTMTzero reagent. Then, Western blotting is performed to detect the S-nitrosylated *MgrA* (SNO-*MgrA*) by using anti-TMT antibody. Total *MgrA* is detected with anti-His antibody. As shown in Fig. R17, the sample in lane 1 was treated with sodium ascorbate, while sample in lane 2 served as the negative control, treated with ultrapure water instead of sodium ascorbate. Western-blotting results showed that S-nitrosylated *MgrA* was only detectable in lane 1, indicating that a proportion of the *MgrA* protein purified from *E. coli* was indeed S-nitrosylated. It was previously reported that the OxyR protein is S-nitrosylated in *E. coli* during anaerobic culture. It is possible that anaerobic conditions may arise in dense overnight cultures of *E. coli*, which induced S-nitrosylation, or that aerobic S-nitrosylation of some proteins may occur in *E. coli* through some previously unrecognized pathway.

Figure R17. Detection of S-nitrosylated MgrA purified from *E. coli*.

I'm surprised that the authors did not make a *walR/mgrA* double mutant to assess the effect of S-nitrosylation on these 2 key regulators. If the aim is to understand the overall impact of NOS on vancomycin susceptibility then this experiment is needed.

As advised by the reviewer, we constructed an *mgrAC12S/walRC67S* double mutant. We found that this double mutant grew as well as the WT strain, which probably because that either *mgrA* or *walR* single mutation did not cause growth defective. Rather, these mutants showed increased growth as compared to the parent strain (Fig. R5a, also see Supplementary Fig. 8a). The MIC of vancomycin was compared between the WT, *mgrAC12S*, *walRC67S*, and *mgrAC12S/walRC67S* strains. We found that the *mgrAC12S/walRC67S* mutant showed growth defects at 4 $\mu\text{g/ml}$ vancomycin, with no significant difference compared to the *mgrAC12S* or *walRC67S* mutant (Fig. R5b, also see Supplementary Fig. 8b), which is consistent with the observation that the Δnos strain retained a MIC of 4 $\mu\text{g/ml}$. This may be because although MgrA and WalR are both transcriptional regulators regulating similar genes involved in autolysin production, they are likely not collaborators, and thus do not have coordinated function and are not dependent on one another to regulate vancomycin resistance^{5,6}.

Besides, several genes, such as *vraTSR*, *graSR* and *rpoB*, were also reported important for VISA development⁷, likely independent of NO-mediated S-nitrosylation. Thus, the function of these genes may also contribute to vancomycin resistance in *S. aureus*. We have added these results in Supplementary Figure 8 and revised the Results section (See Line 298-314) in the manuscript.

Figure R5 (Supplementary Figure 8). Growth curve and vancomycin MICs of the WT and mutant strains.

The discussion was very limited, which is a shame because it would have been

interesting to explore whether NOS could affect susceptibility to other antibiotics (e.g. VISA often have reduced susceptibility to daptomycin) and whether NOS would be a viable therapeutic target to block vancomycin resistance. It would also be useful to understand whether host NO produced by immune cells might affect antibiotic susceptibility.

We thank the reviewer for this constructive suggestion. The occurrence of VISA is usually accompanied by a decrease in sensitivity to daptomycin and teicoplanin²³, we thus also measured the daptomycin or teicoplanin MIC of the WT and Δnos mutant. Similar to the results what we observed from vancomycin, the Δnos mutant showed decreased MIC in the medium containing daptomycin or teicoplanin (Fig. R18; also see Supplementary Fig. 1), suggesting an important and general role of NOS in bacterial antibiotics resistance, which offers NOS a potential treatment target for combating vancomycin resistance. We have added the data in Supplemental Figure 1 and revised the Results section in the manuscript (See Line 106-112).

Figure R18 (Supplementary Figure 1). Daptomycin and teicoplanin MICs of the WT and Δnos mutant strains.

The authors are also grateful for the suggestion to expand our discussion. As advised, we have revised the Discussion section (See Line 316-361) as follows:

In summary, this study demonstrated endogenous S-nitrosylation as a key mechanism for regulation of vancomycin resistance in *S. aureus* and as an important area of future research. VISA use a mechanism involving transcriptional regulators (such as MgrA and WalR) to sense endogenous NO, causing S-nitrosylation and inducing a transcriptional signal to circumvent vancomycin killing (Fig. 7). This study provides novel insights for the clinical treatment of VISA and other antibiotic-resistant pathogens.

MgrA belongs to the MarR family of regulators, which are widely distributed among bacteria and regulate diverse microbial physiological processes. Intriguingly, some other regulators, including SarA and SarZ in *S. aureus* and OhrR in *Bacillus subtilis*, displayed conservation of the redox-sensitive cysteine residue in MgrA (Supplementary Fig. 9). This indicates that S-nitrosylation may affect antibiotic resistance or other cellular functions through a similar mechanism in other bacterial species.

The S-nitrosylated cysteine in MgrA is critical for transcriptional activation of downstream genes that are involved in cell autolysis. S-oxidation of the same cysteine residue has been found to initiate a distinct regulatory signal. It is important to note that

this unique cysteine in MgrA is sometimes phosphorylated²⁴, and that oxidation of Cys12 in the *S. aureus* strain Newman causes antibiotic resistance²⁰. This suggests that *S. aureus* uses modification of Cys12 as a versatile mechanism for responding to cellular nitrosative and oxidative stress and defending against antibiotics. Future studies will uncover other S-nitrosylation-mediated signaling pathways in bacteria.

NOS has also been reported to influence *S. aureus* virulence^{12,13}, sensitivity to oxidative stress^{10,13} and heme stress¹⁴; it also plays essential roles in regulating electron transfer to maintain membrane bioenergetics¹⁵ and modulates aerobic respiratory metabolism and cell physiology¹⁶. Bacterial NOS-derived NO also plays important roles in regulating the formation, swarming, and dispersal of bacterial biofilms^{17,18}. These findings suggest a key role of bacterial NOS or NOS-derived NO in the global regulation of *S. aureus* physiological phenotypes. It is thus worth exploring the mechanisms by which NOS is regulated. The regulatory mechanism mediated by S-nitrosylation may be universal and could be applied to regulation of other physiological phenotypes, and it should therefore be further explored in the future.

Here, we found that the NOS inhibitor L-NAME and vancomycin showed a synergistic antibacterial effect on *S. aureus*. Introducing a NOS inhibitor as a co-treatment with other antibiotics is thus a potential strategy for clinical treatment of severe *S. aureus* infection. However, L-NAME is also an effective inhibitor of mammalian NOS²⁵, and further research is therefore required to discover a bacteria-specific NOS inhibitor.

Host-derived NOS and NO have been suggested to have roles in controlling *S. aureus* infections²⁶⁻³². NO produced by host immune cells has been shown to have broad activity against diverse pathogens and to inhibit *Staphylococcus* virulence^{8,33}. However, it has been found that NO produced by bacteria promotes bacterial virulence and drug resistance¹⁰. Thus, it would be useful to understand whether host cell-produced NO could affect bacterial antibiotic susceptibility. The present study provides a key insight that bacteria may use host-produced NO to resist antibiotic killing during infection. This is a promising direction for future study and potential improvement of antibiotic therapeutic strategies.

Other points:

Line 70. Is NOS produced in all growth phases?

We thank the reviewer for the question. We have measured the transcription of *nos* during different growth phases. The qRT-PCR results showed that *nos* displayed a constant transcription during the whole growth phase, with no more than 2-fold induction at the lag phase (Fig. R19).

Figure R19. Transcription levels of *nos* at different growth phase.

Line81. Please define HTH and explain the typical function of these domains.

As suggested, we have defined HTH and revised the text into ‘This residue was located within the MarR-type helix-turn-helix (HTH) domain, which is responsible for DNA binding, at the interface of the protein dimer’ in the manuscript (See Line 81-82)

Line 82. I think you need to lose the opening ‘In’ from this sentence.

Thanks! We’ve removed the “In” from this sentence. (See Line 83)

Line 88. It would be useful here to explain why you looked at vancomycin susceptibility.

We thank the reviewer’ suggestion. We’ve revised the text as ‘MgrA was reportedly involved in the regulation of vancomycin resistance.’ in the manuscript. (See Line 88)

Line 96-97. I disagree that S-nitrosylation contributes to resistance development since this residue is also present in a vancomycin susceptible strain (8325). I think a more appropriate conclusion is that S-nitrosylation modulates vancomycin susceptibility. On a related note, it would be useful to know whether the C12 residue is conserved across *S. aureus*.

We thank the reviewer’s correction. We’ve revised the text in the manuscript as suggested. The text ‘MgrA S-nitrosylation is important for the development of vancomycin resistance in VISA strain.’ was replaced to ‘MgrA S-nitrosylation is important for the modulation of vancomycin susceptibility’. (See Line 98-99)

To answer the question about the conservation of Cys12 in MgrA, the alignment analysis was performed and showed that Cys12 is conserved among *Staphylococcus* species (Fig. R13).

MgrA ★ C12

```

XN108 1 MSDQHNLKEQLCFSLYNAQRQVNRYYSNKVFKKYNLTYPQFLVLT I LWD 49
Mu50 1 MSDQHNLKEQLCFSLYNAQRQVNRYYSNKVFKKYNLTYPQFLVLT I LWD 49
MW2 1 MSDQHNLKEQLCFSLYNAQRQVNRYYSNKVFKKYNLTYPQFLVLT I LWD 49
N315 1 MSDQHNLKEQLCFSLYNAQRQVNRYYSNKVFKKYNLTYPQFLVLT I LWD 49
NCTC8325 1 MSDQHNLKEQLCFSLYNAQRQVNRYYSNKVFKKYNLTYPQFLVLT I LWD 49

XN108 50 ESPVNVKKVVTEALALDTGTVSPLLKRMEQVDL I KRRERSEVDQREVF I HL 98
Mu50 50 ESPVNVKKVVTEALALDTGTVSPLLKRMEQVDL I KRRERSEVDQREVF I HL 98
MW2 50 ESPVNVKKVVTEALALDTGTVSPLLKRMEQVDL I KRRERSEVDQREVF I HL 98
N315 50 ESPVNVKKVVTEALALDTGTVSPLLKRMEQVDL I KRRERSEVDQREVF I HL 98
NCTC8325 50 ESPVNVKKVVTEALALDTGTVSPLLKRMEQVDL I KRRERSEVDQREVF I HL 98

XN108 99 TDKSET I RPELSNASDKVASASSLSQDEVKELNRL LGKV I HAFDETKEK 147
Mu50 99 TDKSET I RPELSNASDKVASASSLSQDEVKELNRL LGKV I HAFDETKEK 147
MW2 99 TDKSET I RPELSNASDKVASASSLSQDEVKELNRL LGKV I HAFDETKEK 147
N315 99 TDKSET I RPELSNASDKVASASSLSQDEVKELNRL LGKV I HAFDETKEK 147
NCTC8325 99 TDKSET I RPELSNASDKVASASSLSQDEVKELNRL LGKV I HAFDETKEK 147

```

Figure R13. The alignment of MgrA amino acid among different *S. aureus* strains.

Line 127. This is growth inhibition, not killing that is being assayed.

We thank the reviewer for the correction. We've revised the description 'These results indicate that low concentration of NO and the induction of S-nitrosylation on MgrA is required to defend against vancomycin killing.' into 'These results indicated that low-concentration NO and the induction of MgrA S-nitrosylation were required for *S. aureus* vancomycin resistance.' (See Line 146-147)

Line 129. It would be useful to say here that VISA typically have thickened cell walls and explain how this is proposed to confer decreased susceptibility.

We again thank the reviewer for the suggestion! We have revised this sentence into 'VISA strains typically show thickened cell walls, which allow increased capture of vancomycin molecules by free D-Ala-D-Ala residues, protecting the cells from vancomycin killing.' in the text. (See Line 149-151)

Line 137. It is not reasonable to conclude that decreased vancomycin susceptibility is due to a thickened cell wall in the WT relative to mutants. Whilst likely, this is not specifically tested.

We thank the reviewer's suggestion. We've revised the sentence into 'These results demonstrated that endogenous S-nitrosylation of MgrA is important for maintaining uniform cell wall thickness.' in the manuscript. (See Line 157-158)

References:

- 1 Crane, B. R., Sudhamsu, J. & Patel, B. A. Bacterial nitric oxide synthases. *Annual review of biochemistry* **79**, 445-470, doi:10.1146/annurev-biochem-062608-103436 (2010).
- 2 Pattee, P. A. & Neveln, D. S. Transformation analysis of three linkage groups in *Staphylococcus aureus*. *J Bacteriol* **124**, 201-211 (1975).
- 3 Seth, D., Hausladen, A., Wang, Y. J. & Stamler, J. S. Endogenous protein S-Nitrosylation in *E. coli*: regulation by OxyR. *Science* **336**, 470-473, doi:10.1126/science.1215643 (2012).

- 4 Gusarov, I. *et al.* Bacterial nitric-oxide synthases operate without a dedicated redox partner. *The Journal of biological chemistry* **283**, 13140-13147, doi:10.1074/jbc.M710178200 (2008).
- 5 Dubrac, S. & Msadek, T. Identification of genes controlled by the essential YycG/YycF two-component system of *Staphylococcus aureus*. *J Bacteriol* **186**, 1175-1181 (2004).
- 6 Luong, T. T., Dunman, P. M., Murphy, E., Projan, S. J. & Lee, C. Y. Transcription Profiling of the mgrA Regulon in *Staphylococcus aureus*. *Journal of bacteriology* **188**, 1899-1910, doi:10.1128/JB.188.5.1899-1910.2006 (2006).
- 7 Hu, Q., Peng, H. & Rao, X. Molecular Events for Promotion of Vancomycin Resistance in Vancomycin Intermediate *Staphylococcus aureus*. *Front Microbiol* **7**, 1601, doi:10.3389/fmicb.2016.01601 (2016).
- 8 Urbano, R. *et al.* Host Nitric Oxide Disrupts Microbial Cell-to-Cell Communication to Inhibit Staphylococcal Virulence. *Cell Host & Microbe* **23**, 594-+, doi:10.1016/j.chom.2018.04.001 (2018).
- 9 Seth, P. *et al.* Regulation of MicroRNA Machinery and Development by Interspecies S-Nitrosylation. *Cell* **176**, 1014-1025.e1012, doi:https://doi.org/10.1016/j.cell.2019.01.037 (2019).
- 10 Vaish, M. & Singh, V. K. Antioxidant Functions of Nitric Oxide Synthase in a Methicillin Sensitive *Staphylococcus aureus*. *Int J Microbiol* **2013**, 312146, doi:10.1155/2013/312146 (2013).
- 11 Sapp, A. M. *et al.* Contribution of the nos-pdt operon to virulence phenotypes in methicillin-sensitive *Staphylococcus aureus*. *PLoS one* **9**, e108868, doi:10.1371/journal.pone.0108868 (2014).
- 12 James, K. L. *et al.* Interplay of Nitric Oxide Synthase (NOS) and SrrAB in Modulation of *Staphylococcus aureus* Metabolism and Virulence. *Infect Immun* **87**, doi:10.1128/iai.00570-18 (2019).
- 13 van Sorge, N. M. *et al.* Methicillin-resistant *Staphylococcus aureus* bacterial nitric-oxide synthase affects antibiotic sensitivity and skin abscess development. *The Journal of biological chemistry* **288**, 6417-6426, doi:10.1074/jbc.M112.448738 (2013).
- 14 Surdel, M. C., Dutter, B. F., Sulikowski, G. A. & Skaar, E. P. Bacterial Nitric Oxide Synthase Is Required for the *Staphylococcus aureus* Response to Heme Stress. *ACS infectious diseases* **2**, 572-578, doi:10.1021/acsinfecdis.6b00081 (2016).
- 15 Kinkel, T. L. *et al.* An essential role for bacterial nitric oxide synthase in *Staphylococcus aureus* electron transfer and colonization. *Nature Microbiology* **2**, doi:10.1038/nmicrobiol.2016.224 (2017).
- 16 Mogen, A. B. *et al.* *Staphylococcus aureus* nitric oxide synthase (saNOS) modulates aerobic respiratory metabolism and cell physiology. *Mol Microbiol* **105**, 139-157, doi:10.1111/mmi.13693 (2017).
- 17 Vasilieva, S. V. & Streltsova, D. A. Interaction of messengers—endogenous NO and H₂S gasotransmitters—in signaling and regulatory processes in bacterial cells. *Doklady Biochemistry and Biophysics* **461**, 114-118, doi:10.1134/S1607672915020131 (2015).
- 18 Schreiber, F. *et al.* The role of nitric-oxide-synthase-derived nitric oxide in multicellular traits of *Bacillus subtilis* 3610: biofilm formation, swarming, and dispersal. *BMC Microbiol*

- 11**, 111, doi:10.1186/1471-2180-11-111 (2011).
- 19 Trotonda, María P., Xiong, Yan Q., Memmi, G., Bayer, Arnold S. & Cheung, Ambrose L. Role of mgrA and sarA in Methicillin-Resistant Staphylococcus aureus Autolysis and Resistance to Cell Wall-Active Antibiotics. *The Journal of infectious diseases* **199**, 209-218, doi:10.1086/595740 (2009).
- 20 Chen, P. R. *et al.* An oxidation-sensing mechanism is used by the global regulator MgrA in Staphylococcus aureus. *Nature Chemical Biology* **2**, 591-595, doi:10.1038/nchembio820 (2006).
- 21 Bose, J. L., Lehman, M. K., Fey, P. D. & Bayles, K. W. Contribution of the Staphylococcus aureus Atl AM and GL murein hydrolase activities in cell division, autolysis, and biofilm formation. *PLoS one* **7**, e42244, doi:10.1371/journal.pone.0042244 (2012).
- 22 Tiwari, K. B., Gatto, C., Walker, S. & Wilkinson, B. J. Exposure of Staphylococcus aureus to Targocil Blocks Translocation of the Major Autolysin Atl across the Membrane, Resulting in a Significant Decrease in Autolysis. *Antimicrob Agents Chemother* **62**, doi:10.1128/aac.00323-18 (2018).
- 23 Wang, W. & Sun, B. VraCP regulates cell wall metabolism and antibiotic resistance in vancomycin-intermediate Staphylococcus aureus strain Mu50. *The Journal of antimicrobial chemotherapy* **76**, 1712-1723, doi:10.1093/jac/dkab113 (2021).
- 24 Sun, F. *et al.* Protein cysteine phosphorylation of SarA/MgrA family transcriptional regulators mediates bacterial virulence and antibiotic resistance. *Proc Natl Acad Sci U S A* **109**, 15461-15466, doi:10.1073/pnas.1205952109 (2012).
- 25 Strijdom, H., Muller, C. & Lochner, A. Direct intracellular nitric oxide detection in isolated adult cardiomyocytes: flow cytometric analysis using the fluorescent probe, diaminofluorescein. *Journal of Molecular and Cellular Cardiology* **37**, 897-902, doi:https://doi.org/10.1016/j.yjmcc.2004.05.018 (2004).
- 26 Li, C. *et al.* Interleukin-33 increases antibacterial defense by activation of inducible nitric oxide synthase in skin. *PLoS pathogens* **10**, e1003918, doi:10.1371/journal.ppat.1003918 (2014).
- 27 McInnes, I. B., Leung, B., Wei, X. Q., Gemmell, C. C. & Liew, F. Y. Septic arthritis following Staphylococcus aureus infection in mice lacking inducible nitric oxide synthase. *J Immunol* **160**, 308-315 (1998).
- 28 Richardson, A. R., Libby, S. J. & Fang, F. C. A nitric oxide-inducible lactate dehydrogenase enables Staphylococcus aureus to resist innate immunity. *Science* **319**, 1672-1676, doi:10.1126/science.1155207 (2008).
- 29 Rothfork, J. M. *et al.* Inactivation of a bacterial virulence pheromone by phagocyte-derived oxidants: new role for the NADPH oxidase in host defense. *Proc Natl Acad Sci U S A* **101**, 13867-13872, doi:10.1073/pnas.0402996101 (2004).
- 30 Sasaki, S. *et al.* Protective role of nitric oxide in Staphylococcus aureus infection in mice. *Infect Immun* **66**, 1017-1022, doi:10.1128/IAI.66.3.1017-1022.1998 (1998).
- 31 Urbano, R. *et al.* Host Nitric Oxide Disrupts Microbial Cell-to-Cell Communication to Inhibit Staphylococcal Virulence. *Cell Host Microbe* **23**, 594-606 e597, doi:10.1016/j.chom.2018.04.001 (2018).
- 32 Kinkel, T. L. *et al.* An essential role for bacterial nitric oxide synthase in Staphylococcus aureus electron transfer and colonization. *Nat Microbiol* **2**, 16224,

doi:10.1038/nmicrobiol.2016.224 (2016).

- 33 Fang, F. C. Antimicrobial reactive oxygen and nitrogen species: concepts and controversies. *Nat Rev Microbiol* **2**, 820-832, doi:10.1038/nrmicro1004 (2004).

Reviewers' comments:

Reviewer #1 (Remarks to the Author):

The three people who reviewed the previous version of this paper all expressed appreciation of the novelty of the ideas presented, but all of us wanted to see supporting data from control or additional experiments. The authors have considered all of the suggestions made and indeed presented many additional results in response to our questions. Unfortunately none of the additional data strongly support the hypothesis presented. We are left with the same concerns raised in response to the previous draft of the paper. The 25-page response to comments by the reviewers contain no strong supporting evidence that the claims made are correct - or physiologically relevant. The authors repeatedly state the well-known fact that nitric oxide synthase (Nos) requires oxygen to function. The one additional result consistent with expectation is that any vancomycin resistance during anaerobic growth is not dependent upon Nos. However, *S. aureus* has multiple enzymes that catalyse NO production from nitrite. If the authors are correct, one would expect the transcription factors that regulate cell wall degradation to be strongly nitrosylated during anaerobic growth in the presence of nitrate or nitrite (which is toxic at concentrations above about 0.1 mM), but not in the presence of fumarate or glucose. No such difference was found by the authors, possibly because of inadequate experimental design. The double mutant was not more resistant than the parent or single mutants. An additive difference would have been supporting evidence, but the negative result reported proves nothing. The authors try to explain why quantitative data on the extent of nitration of individual proteins cannot be provided. They also explain why NO-saturated water was not used, but the excuse is weak. Anaerobic jars could – should – have been used.

Reviewer #2 (Remarks to the Author):

The authors have performed a series of experiments to address my (and the other reviewers') concerns. I still believe that the manuscript would significantly benefit from more direct evidence for the importance of nitrosylation of MgrA at a specific position with that nitrosylation being demonstrated by methods such as MS (e.g. one could tag MgrA by His-Tag or otherwise, purify from WT and nos mutant, and subject to MS/MS analysis). However, in light of the overall very responsive resubmission I am happy with the manuscript as it stands.

Reviewer #3 (Remarks to the Author):

The authors have undertaken significant additional experimental work that fully addresses my concerns.

Our point-by-point response to the reviewers' comments

Reviewer #1 (Remarks to the Author):

The three people who reviewed the previous version of this paper all expressed appreciation of the novelty of the ideas presented, but all of us wanted to see supporting data from control or additional experiments. The authors have considered all of the suggestions made and indeed presented many additional results in response to our questions. Unfortunately, none of the additional data strongly support the hypothesis presented. We are left with the same concerns raised in response to the previous draft of the paper. The 25-page response to comments by the reviewers contain no strong supporting evidence that the claims made are correct - or physiologically relevant.

Response: We very much appreciated that the reviewer has provided us with very good suggestions for better improvement of this study. We are so sorry that our additional data did not meet the reviewer's requirements. Some of which are due to technical problems, such as the anaerobic experiments, for which we were very thankful that the reviewer has provided us with important suggestions in the second round of review. We have conducted more control experiments and obtained important supporting data (showed below), such as the effect of different concentrations of nitrate, nitrite or fumarate on vancomycin resistance in another more suitable medium under anaerobic conditions. The S-nitrosylation of MgrA or WalR in the XN108 WT and Δnos strain and the extent of S-nitrosylated MgrA have also been further measured. Data were shown below.

The authors repeatedly state the well-known fact that nitric oxide synthase (Nos) requires oxygen to function. The one additional result consistent with expectation is that any vancomycin resistance during anaerobic growth is not dependent upon Nos. However, *S. aureus* has multiple enzymes that catalyze NO production from nitrite. If the authors are correct, one would expect the transcription factors that regulate cell wall degradation to be strongly nitrosylated during anaerobic growth in the presence of nitrate or nitrite (which is toxic at concentrations above about 0.1 mM), but not in the presence of fumarate or glucose. No such difference was found by the authors, possibly because of inadequate experimental design.

Response: We thank the reviewer's kind reminding and good suggestion. As reminded by the reviewer, no promotion effect of nitrate or nitrite was observed on the vancomycin MICs of *S. aureus* in anaerobic cultures, which may be due to some technical problems and inadequate experimental design, such as excessive salt concentrations or inappropriate culture medium. Besides, it may be difficult to cause

an exponential increase in MIC by just adding salts to the medium at the beginning, because a constant and stable concentration of salts cannot be guaranteed. Thus, we evaluated the effect of these salts on vancomycin resistance by examining their effect on growth under vancomycin stress. Anaerobiosis was reported to bring about changes in nutritional requirements and metabolic pathways of staphylococci, and BHI (Brain Heart Infusion Broth) has been used to support better anaerobic growth of staphylococci than MH (Mueller-Hinton) we previously used². Therefore, we detected the effect of these salts with different concentrations on growth in BHI containing 5 µg/ml vancomycin in anaerobic jars. As shown in Fig. R1, the decreased growth caused by vancomycin could be greatly promoted by 10 mM nitrate, while neither nitrite nor fumarate showed no promotion effect but exhibited some inhibitory effect, indicating that nitrate-mediated NO production may have the capacity to promote the resistance of *S. aureus* to vancomycin. A similar promotion effect of nitrate on vancomycin-induced decreased growth was also observed in PN medium (Fig. R2), which is consistent with the reviewer's idea that, in addition to NOS under aerobic condition, NO generated in *S. aureus* from nitrate under anaerobic condition may also be used to play a similar regulatory role in vancomycin resistance.

As reported, nitrate-generated NO in *E. coli* has been proved to mediate S-nitrosylation of bacterial proteins¹, which in turn are involved in the regulation of diverse aspects of cellular function, such as resistance to nitrosative or oxidative stress to maintain anaerobic growth^{1,3} and bacterial motility⁴. Therefore, we thank very much of the reviewer's suggestion and strongly agree with the reviewer's point that *S. aureus* may also has the ability to use nitrate-generated NO to mediate S-nitrosylation of *S. aureus* proteins, which may be involved in the regulation of antibiotic resistance under anaerobic conditions, which is consistent with the phenomenon we observed above. We will further study the underlying mechanism of how vancomycin resistance of *S. aureus* is regulated during anaerobic growth in the future.

Figure R1. Growth curve of the XN108 WT strain in BHI medium containing 5 $\mu\text{g/ml}$ vancomycin when added with different concentrations of nitrate (a-c), nitrite (d-f) or fumarate (g-i). V0 or V5 represents BHI added with 0 or 5 $\mu\text{g/ml}$ vancomycin respectively. Nitra-10, Nitra-1 or Nitra-0.1 represents BHI added with 10, 1 or 0.1 mM nitrate respectively. Nitri-10, Nitri-1 or Nitri-0.1 represents BHI added with 10, 1 or 0.1 mM nitrite respectively. Fum-10, Fum-1 or Fum-0.1 represents BHI added with 10, 1 or 0.1 mM fumarate respectively.

Figure R2. Growth curve of the XN108 WT strain in PN medium containing 2 µg/ml vancomycin when added with different concentrations of nitrate (a-c), nitrite (d-f) or fumarate (g-i). V0 or V2 represents PN added with 0 or 2 µg/ml vancomycin respectively. Nitra-10, Nitra-1 or Nitra-0.1 represents PN added with 10, 1 or 0.1 mM nitrate respectively. Nitri-1, Nitri-0.1 or Nitri-0.02 represents PN added with 1, 0.1 or 0.02 mM nitrite respectively. Fum-10, Fum-1 or Fum-0.1 represents PN added with 10, 1 or 0.1 mM fumarate respectively.

The double mutant was not more resistant than the parent or single mutants. An additive difference would have been supporting evidence, but the negative result reported proves nothing.

Response: We thank very much of the reviewer for the question and discussion. Our results showed that the double mutant strain *mgrAC12S/walRC67S* exhibited similar vancomycin MIC compared to the *mgrAC12S* or *walRC67S* mutant, which is consistent with the observation that the Δnos strain, which lost the ability to generate NO during aerobic growth, also retained the same MIC. Either *mgrAC12S* or *walRC67S* mutant caused a significant decrease in vancomycin resistance, suggesting important role of NO-induced S-nitrosylation of these transcriptional regulators in controlling vancomycin resistance of VISA, which is the main conclusion of this study. Whether these two regulators will coordinate together to modulate vancomycin resistance is unclear and requires further investigation. Besides, there are also many

other genes, such as other two-component systems (e.g. *vraTSR* and *graSR*), small transcription regulators (e.g. *sarA*) or some functional genes (e.g. *rpoB* and *stpI*), that are involved in vancomycin resistance of VISA^{5,6}, likely independent of NO-mediated S-nitrosylation. Therefore, *mgrA* or *walR* may not have to collaborate to regulate resistance and the double mutant may not result in further decrease in resistance.

The authors try to explain why quantitative data on the extent of nitration of individual proteins cannot be provided.

Response: We are very sorry for this and agree with the reviewer that it is important to provide such a critical data for this work. Therefore, we detected the extent of S-nitrosylation of MgrA by using His-tagged MgrA (MgrA-His) purified from the XN108 WT strain. An S-Nitrosylation Western Blot Kit (90105, Pierce) was used. In this assay, the unmodified cysteines of two samples of purified His-tagged MgrA with equal concentration were firstly blocked with or without a sulfhydryl-reactive compound MMTS (Methyl Methanethiosulfonate). S-nitrosylated cysteines were then selectively reduced with sodium ascorbate in HENS buffer for specific and irreversible labeling with iodoTMTzero reagents. Then, Western blot was performed to detect the S-nitrosylated MgrA (MgrA-SNO) by using an anti-TMT antibody. Total MgrA was detected with an anti-His antibody. As shown in Fig. R3, to better assess the extent of modifications accounted for, samples with gradient concentrations were analyzed with Western blot. Quantitative analysis of the gray scale revealed that the amount of S-nitrosylated MgrA without MMTS block was more than seven times higher than that of the MMTS blocked sample. Indicating that the extent of S-nitrosylated MgrA of all the copies of His-tagged MgrA purified from the WT strain should be less than 14.3%.

We also measured the ratio of S-nitrosylation of endogenous proteins with His-tag purified from the XN108 WT and Δnos strain by using the S-Nitrosylation Western Blot Kit. As shown in Fig. R4a (also see Fig. 1f), S-nitrosylated MgrA was obviously detected in the WT strain rather than in the Δnos strain. A similar result was also observed in the S-nitrosylated WalR (Fig. R4b; also see Supplementary Fig. 8). These results indicate that S-nitrosylation formation of MgrA or WalR is dependent on the NO generated from NOS. As the principle of the S-Nitrosylation Western blot kit is similar to that of the kits that we used to identify S-nitrosylated cysteines by MS, so we think that the Western blot results we give here can provide similar direct evidence for the importance of the NOS-dependent S-nitrosylation of MgrA or WalR as that a MS can provide. We have added the WB result of MgrA in Figure 1f and the WB result of WalR in Supplementary Figure 8, and revised the Results section in the manuscript (See Line 85-90 and line 282-286).

Figure R3. Detection of S-nitrosylation of His-tagged MgrA purified from the XN108 WT strain after blocking without or with MMTS. Concentrations of these samples were firstly adjusted to be consistent and noted as 1, and then, diluted samples were analyzed with Western blot. The dilution times were marked on the top of the picture.

Figure R4. Detection of S-nitrosylated MgrA (a) or WalR (b) purified from the XN108 WT and Δnos strain.

They also explain why NO-saturated water was not used, but the excuse is weak. Anaerobic jars could – should – have been used.

Response: We thank the reviewer's suggestion and very sorry for the misunderstanding of the question as mentioned, "The authors cannot use NO-saturated water", in the first review. We strongly agree with the reviewer that it is very helpful to determine the effect of NO itself on bacteria through direct injection of NO into the anaerobic incubator or through NO-saturated water. We reviewed the literature and found that NO-saturated water can be prepared by bubbling NO gas, that has been purified from higher oxides by passing it through a 1 M solution of KOH, into water in an airtight device with equipment to collect and detoxify NO gas, as NO is toxic and slightly soluble in water⁷. Immediately before the reaction, the NO concentration should be measured and adjusted. We have used anaerobic jars to perform the above growth experiments. However, we are sorry that we do not have the ability to prepare the NO-saturated water because we do not have such an airtight device with suitable equipment to collect and detoxify NO gas. As reported, NO donors are widely used in studying the NO-mediated modification and regulation⁸⁻¹¹ under aerobic condition, indicating that NO donor can satisfy most of the aerobic experimental purposes. Therefore, NO donor was used in our study to explore the effect of NO on the phenotypes involved in vancomycin resistance.

Reviewer #2 (Remarks to the Author):

The authors have performed a series of experiments to address my (and the other reviewers') concerns. I still believe that the manuscript would significantly benefit from more direct evidence for the importance of nitrosylation of MgrA at a specific position with that nitrosylation being demonstrated by methods such as MS (e.g. one could tag MgrA by His-Tag or otherwise, purify from WT and *nos* mutant, and subject to MS/MS analysis). However, in light of the overall very responsive resubmission I am happy with the manuscript as it stands.

Response: We thank and agree with the reviewer's suggestion that tagged protein can be used to give more direct evidence for the importance of S-nitrosylation of MgrA at a specific position, which is important to improve and provide critical data for this work. As suggested, we detected the S-nitrosylation of His-tagged MgrA (MgrA-His) purified from the XN108 WT or Δnos strain using an S-Nitrosylation Western Blot Kit (90105, Pierce). In this assay, the unmodified cysteines were firstly blocked with a sulfhydryl-reactive compound MMTS (Methyl Methanethiosulfonate). S-nitrosylated cysteines were then selectively reduced with sodium ascorbate in HENS buffer for specific and irreversible labeling with iodoTMTzero reagents. Then, Western blot was performed to detect the S-nitrosylated MgrA (MgrA-SNO) by using an anti-TMT antibody. Total MgrA was detected with an anti-His antibody. As shown in Fig. R4a (also see Fig. 1f), S-nitrosylated MgrA was obviously detected in the WT strain rather than in the Δnos strain. A similar result was also observed in the S-nitrosylated WalR (Fig. R4b; also see Supplementary Fig. 8). These results indicate that the S-nitrosylation formation of MgrA or WalR is dependent on the NO generated from NOS. As the principle of the S-Nitrosylation Western blot kit is similar to that of the kits that we used to identify S-nitrosylated cysteines by MS, we think that the Western blot results we give here can provide similar direct evidence for the importance of the NOS-dependent S-nitrosylation of MgrA or WalR as that a MS can provide. We have added the WB result of MgrA in Figure 1f and the WB result of WalR in Supplementary Figure 8, and revised the Results section in the manuscript (See Line 85-90 and line 282-286).

Figure R4. Detection of S-nitrosylated MgrA (a) or WalR (b) purified from the XN108 WT and Δnos strain.

We also estimated the extent of S-nitrosylated MgrA of all the copies of His-tagged MgrA purified from the XN108 WT strain by using the S-Nitrosylation Western Blot Kit. Two samples of purified His-tagged MgrA with equal concentration were blocked with or without MMTS, and then both of them were reduced and labeled with iodoTMTzero reagents. TMT-labeled samples with equal concentration were analyzed using Western blot to detect the S-nitrosylated MgrA. As shown in Fig. R3, to better assess the extent of modifications accounted for, samples with gradient concentrations were analyzed with Western blot. The quantitative analysis of the gray scale revealed that the amount of S-nitrosylated MgrA without MMTS block was more than seven times higher than that of the MMTS blocked sample. Indicating that the extent of S-nitrosylated MgrA of all the copies of His-tagged MgrA purified from the WT strain should be less than 14.3%.

Figure R3. Detection of S-nitrosylation of His-tagged MgrA purified from the XN108 WT strain after blocking without or with MMTS. Concentrations of these samples were firstly adjusted to be consistent and noted as 1, and then, diluted samples were analyzed with Western blot. The dilution times were marked on the top of the picture.

Reviewer #3 (Remarks to the Author):

The authors have undertaken significant additional experimental work that fully addresses my concerns.

Response: We thank very much of the reviewer for the support of our work.

References:

- 1 Seth, D., Hausladen, A., Wang, Y. J. & Stamler, J. S. Endogenous protein S-Nitrosylation in *E. coli*: regulation by OxyR. *Science* **336**, 470-473, doi:10.1126/science.1215643 (2012).
- 2 Harrell, L. J. & Evans, J. B. Effect of anaerobiosis on antimicrobial susceptibility of staphylococci. *Antimicrob Agents Chemother* **11**, 1077-1078, doi:10.1128/aac.11.6.1077 (1977).
- 3 Barraud, N. *et al.* Lifestyle-specific S-nitrosylation of protein cysteine thiols regulates *Escherichia coli* biofilm formation and resistance to oxidative stress. *NPJ Biofilms Microbiomes* **7**, 34, doi:10.1038/s41522-021-00203-w (2021).
- 4 Seth, D. *et al.* A Multiplex Enzymatic Machinery for Cellular Protein S-nitrosylation. *Mol Cell* **69**, 451-464.e456, doi:10.1016/j.molcel.2017.12.025 (2018).
- 5 Hu, Q., Peng, H. & Rao, X. Molecular Events for Promotion of Vancomycin Resistance in Vancomycin Intermediate *Staphylococcus aureus*. *Front Microbiol* **7**, 1601, doi:10.3389/fmicb.2016.01601 (2016).
- 6 Manna, A. C., Ingavale, S. S., Maloney, M., van Wamel, W. & Cheung, A. L. Identification of sarV (SA2062), a new transcriptional regulator, is repressed by SarA and MgrA (SA0641) and involved in the regulation of autolysis in *Staphylococcus aureus*. *J Bacteriol* **186**, 5267-5280, doi:10.1128/JB.186.16.5267-5280.2004 (2004).
- 7 Gusarov, I. & Nudler, E. NO-mediated cytoprotection: Instant adaptation to oxidative stress in bacteria. *Proc. Natl. Acad. Sci. U. S. A.* **102**, 13855-13860, doi:10.1073/pnas.0504307102 (2005).
- 8 Urbano, R. *et al.* Host Nitric Oxide Disrupts Microbial Cell-to-Cell Communication to Inhibit Staphylococcal Virulence. *Cell Host Microbe* **23**, 594-+, doi:10.1016/j.chom.2018.04.001 (2018).
- 9 Richardson, A. R., Libby, S. J. & Fang, F. C. A nitric oxide-inducible lactate dehydrogenase enables *Staphylococcus aureus* to resist innate immunity. *Science* **319**, 1672-1676, doi:10.1126/science.1155207 (2008).
- 10 Seth, P. *et al.* Regulation of MicroRNA Machinery and Development by Interspecies S-Nitrosylation. *Cell* **176**, 1014-1025.e1012, doi:https://doi.org/10.1016/j.cell.2019.01.037 (2019).
- 11 Gusarov, I. *et al.* Bacterial Nitric Oxide Extends the Lifespan of *C. elegans*. *Cell* **152**, 818-830, doi:https://doi.org/10.1016/j.cell.2012.12.043 (2013).

REVIEWERS' COMMENTS

Reviewer #1 (Remarks to the Author):

All three reviewers of the previous version of this paper were concerned about the reproducibility of the data, the statistical significance of differences reported, and whether the effects were direct or indirect consequences of S-nitrosylation of transcription factors involved in the regulation of cell wall turnover. The authors have now included substantial extra data to answer these criticisms. Reviewers 2 and 3 were immediately satisfied with the extensive response of the authors. I was less convinced, mainly because of controversial statements in the long "response to reviewers". The response included a figure showing the effects of two variables on growth, but an important control was not shown. I therefore cross-referenced the figures and the text. No obvious fault was found: the text accurately describes data presented in the figures. I therefore recommend that the current version of the paper can be accepted for publication.

Point-by-point response to the reviewers' comments

Reviewer #1 (Remarks to the Author):

All three reviewers of the previous version of this paper were concerned about the reproducibility of the data, the statistical significance of differences reported, and whether the effects were direct or indirect consequences of S-nitrosylation of transcription factors involved in the regulation of cell wall turnover. The authors have now included substantial extra data to answer these criticisms. Reviewers 2 and 3 were immediately satisfied with the extensive response of the authors. I was less convinced, mainly because of controversial statements in the long “response to reviewers”. The response included a figure showing the effects of two variables on growth, but an important control was not shown. I therefore cross-referenced the figures and the text. No obvious fault was found: the text accurately describes data presented in the figures. I therefore recommend that the current version of the paper can be accepted for publication.

Response:

We thank very much for the reviewer's support of our work!